# mTORC1 activation in lung mesenchyme drives sex- and age-dependent pulmonary structure and function decline

Kseniya Obraztsova[1,2], Maria C. Basil[1,2], Ryan Rue[1], Aravind Sivakumar[3], Susan M. Lin[1], Alexander R. Mukhitov [1], Andrei I. Gritsiuta [1], Jilly F. Evans[1], Meghan Kopp[1], Jeremy Katzen[1,2], Annette Robichaud[4], Elena N. Atochina-Vasserman [1], Shanru Li[2,5], Justine Carl[2,5], Apoorva Babu[2,5], Michael P. Morley[2,5], Edward Cantu[2,6], Michael F. Beers[1,2], David B. Frank [2,3,5], Edward E. Morrisey[2,5] & Vera P. Krymskaya [1,2,5]✉

Lymphangioleiomyomatosis (LAM) is a rare fatal cystic lung disease due to bi-allelic inactivating mutations in *tuberous sclerosis complex (TSC1/TSC2)* genes coding for suppressors of the mechanistic target of rapamycin complex 1 (mTORC1). The origin of LAM cells is still unknown. Here, we profile a LAM lung compared to an age- and sex-matched healthy control lung as a hypothesis-generating approach to identify cell subtypes that are specific to LAM. Our single-cell RNA sequencing (scRNA-seq) analysis reveals novel mesenchymal and transitional alveolar epithelial states unique to LAM lung. This analysis identifies a mesenchymal cell hub coordinating the LAM disease phenotype. Mesenchymal-restricted deletion of *Tsc2* in the mouse lung produces a mTORC1-driven pulmonary phenotype, with a progressive disruption of alveolar structure, a decline in pulmonary function, increase of rapamycin-sensitive expression of WNT ligands, and profound female-specific changes in mesenchymal and epithelial lung cell gene expression. Genetic inactivation of WNT signaling reverses age-dependent changes of mTORC1-driven lung phenotype, but WNT activation alone in lung mesenchyme is not sufficient for the development of mouse LAM-like phenotype. The alterations in gene expression are driven by distinctive crosstalk between mesenchymal and epithelial subsets of cells observed in mesenchymal *Tsc2*-deficient lungs. This study identifies sex- and age-specific gene changes in the mTORC1-activated lung mesenchyme and establishes the importance of the WNT signaling pathway in the mTORC1-driven lung phenotype.

---

[1] Division of Pulmonary, Allergy, and Critical Care Medicine, Department of Medicine, University of Pennsylvania, Philadelphia, PA, USA. [2] Lung Biology Institute, University of Pennsylvania, Philadelphia, PA, USA. [3] Children Hospital of Philadelphia, Philadelphia, PA, USA. [4] SCIREQ Inc., Montreal, QC, Canada. [5] Cardiovascular Institute, University of Pennsylvania, Philadelphia, PA, USA. [6] Department of Surgery, Perelman School of Medicine, University of Pennsylvania, Philadelphia, PA, USA. ✉email: krymskay@pennmedicine.upenn.edu

Lung function is controlled by complex interactions between a wide variety of cell types. Defects in this well-orchestrated cellular crosstalk can result in impairment of lung function and a variety of pulmonary pathologies. A subgroup of lung diseases is sex/gender- and age-dependent such as pulmonary arterial hypertension which occurs disproportionately in women[1]. Gender also affects the susceptibility and severity of some lung diseases[1]. The rare genetic disease pulmonary lymphangioleiomyomatosis (LAM) occurs almost exclusively in women of childbearing age due to loss of function of the tumor suppressor gene *tuberous sclerosis complex 2* (*TSC2*), a negative regulator of the mechanistic target of rapamycin complex 1 (mTORC1)[2,3]. Nearly all reported symptomatic LAM cases are in women[4], and pregnancy exacerbates LAM[5]. The study of the monogenic disease LAM may uncover links between female-specific lung pathologies and the activation of the mTORC1, a key controller of cell growth, metabolism, and aging with a well-established role in cancer and stem cell biology[6,7].

LAM is a disease characterized by cystic airspace enlargement, focal proliferative nests of smooth-muscle-like cells, and progressive pulmonary function decline, often requiring lung transplantation[4]. LAM can arise sporadically or from somatic mutations with bi-allelic *TSC1* or *TSC2* gene inactivation in LAM associated with the Tuberous Sclerosis Complex[3,8]. After an outstanding collaboration between clinicians, scientists, and patients, Sirolimus (rapamycin) was shown to halt the progression of LAM symptoms in a subset of LAM patients and is now the only FDA-approved drug for the treatment of LAM[4,9]. Although beneficial, in many LAM patients sirolimus (or close analogs named rapalogs) are not curative, and after cessation of therapy, the disease progresses[10,11]. Also, some LAM patients are intolerant of, or unresponsive to, rapalogs[4,12]. In contrast to lung cancers, where most tumors are discrete masses surrounded by normal lung parenchyma, wide-spread cystic lung changes are observed in LAM. It is not well understood how a subset of TSC1/2-null cells drives the observed pathological changes throughout the whole LAM lung. A major limitation in defining critical signaling mechanisms in LAM and in identifying new therapeutics has been the lack of a relevant genetic animal model. Homozygous $Tsc1^{-/-}$ and $Tsc2^{-/-}$ mice are embryonic lethal and spontaneous tumors in heterozygous $Tsc1^{+/-}$ and $Tsc2^{+/-}$ mice are mainly kidney cystadenomas and liver hemangiomas[13]. The spontaneous occurrence of lung tumors in heterozygous $Tsc1^{+/-}$ and $Tsc2^{+/-}$ mice, or in Eker rats with a naturally occurring *Tsc2* mutation, is extremely rare and occurs without major changes of lung parenchyma and only in very aged animals[13].

To address these challenges and to generate a hypothesis about the cell of origin of LAM, we perform single-cell RNA sequencing (scRNA-seq) of a LAM lung and compare this with an age-and sex-matched normal human lung. Our analysis identifies a unique LAM lung cell cluster and novel mesenchymal and alveolar epithelium transitional cell states in LAM lung. To elucidate the role of the abnormal lung mesenchyme, we generate a mTORC1 gain-of-function mouse model by deleting *Tsc2* specifically in lung mesenchyme progenitor cells. mTORC1 activation in lung mesenchyme induces upregulation of select WNT ligands only in females and produces a significant decline in lung function and pathological structural changes in lung parenchyma that is exacerbated by pregnancies. Our study demonstrates that a lung alveolar mesenchymal hub of cells coordinates novel cellular crosstalk for the development of a LAM-like phenotype. Importantly, the upregulation of WNT signaling alone in the lung mesenchyme is not sufficient for the development of LAM-like changes, but genetic ablation of WNT signaling in lung mesenchyme prevents age-dependent airspace enlargement in the mTORC1 gain-of-function mouse model. Thus, our study demonstrates a critical role for *Tsc2*-dependent mTORC1 and WNT signaling pathways in the regulation of the lung structure and function.

## Results

**LAM lung contains transitional mesenchymal and alveolar epithelial cell states.** The constitutive activation of mTORC1 in LAM lesions causes cystic airspace enlargement (Fig. 1a) leading to spontaneous pneumothoraxes and progressive, accelerated loss of pulmonary function. Typical LAM lesions consist of a nest of cells formed by "immature"-looking mesenchymal smooth-muscle-like cells expressing smooth-muscle α-actin (SMA) and positive for phospho-ribosomal protein S6 (pS6), a marker of mTORC1 activation[2–4] (Fig. 1b). LAM lesions also stain positively for melanocytic markers detected with HMB45[3,4], chemoattractant proteins CCL2 and CXCL12[14] the lymphangiogenic factor VEGFD[15], and also contain activated fibroblasts[16]. However, the origin or the evolution of the "bona fide" cell of disease origin in LAM is hotly debated[17–19].

To identify differences between LAM and healthy lung on a single-cell level, we performed scRNA-seq of cells dissociated from LAM lung and age- and sex-matched normal human control lung parenchyma. Four major lung cell clusters, namely epithelial, endothelial, mesenchymal, and immune cells, were identified both in control and LAM lung parenchyma (Fig. 2a). Comparative analysis of these core cell types revealed distinctive cellular states in LAM lung epithelial and mesenchymal clusters (Fig. 2b, c). In control lung, the alveolar epithelial cell (AEC) niche was divided into two canonical clusters: alveolar type 2 (AT2), marked by the high differential expression of *SFTPC, SFTPD,* and *SFTPA* genes, and alveolar type-1 (AT1) cells, characterized by *HOPX, AGER,* and *AQP4* gene expression (Fig. 2b, d, Supplementary Fig. 1A, B). In contrast, LAM lung contained a third distinct AEC sub-cluster defined by high differential expression of both AT2 and AT1 cell markers, including *SFTPC, SFTPD, AGER, HOPX,* and *AQP4* (Fig. 2c, d, Supplementary Fig. 1C, D), suggesting it to be a transitional state of AT2 to AT1 cells (AT2/AT1 cells). To verify this finding, we located this cell type in the lung sections of LAM patients using immunofluorescent double staining for human highly specific AT1 and AT2 markers (HTI-56[20] and HTII-280[21], respectively) and demonstrated, that alveolar regions in LAM lung contain a statistically significantly increased number of AT2 cells expressing AT1 cell markers, compared to the age- and sex-matched healthy controls (Fig. 2e, f).

Next, we examined the mesenchymal cell populations of the lung parenchyma (airways, bronchus, and large blood vessels were dissected out from the fresh tissue). Both the control and LAM lung contained mesenchymal alveolar cells (MACs), defined by the high differential expression of *PDGFRa, DCN,* and *SFRP4* genes (Supplementary Fig. 2). Importantly, only LAM lung contained an additional mesenchymal subtype, which we identified as "fibrotically activated MACs" based on the marked expression of pro-fibrotic gene markers *COL3A1, COL1A1,* and *ACTA2* (Fig. 2c, d, and Supplementary Fig. 2C, D). Another specific mesenchymal subtype found in LAM lung we identified as "differentiated smooth muscle cells" (dSMCs), based on the enhanced expression of differentiated SMC gene markers, including *MYLK, MYL9, DES,* but lacking *PDGFRβ* expression, a marker of alveolar myofibroblasts (AMFs) (Fig. 2c, d, and Supplementary Fig. 2C, D). These findings suggest that fibrotically activated MACs, and a subset of dSMCs, may constitute SMA-positive cell clusters in the lesions of the LAM lung.

The most distinctive cellular subset, which we named "LAM lung cells", was found specifically in LAM lung, and did not have

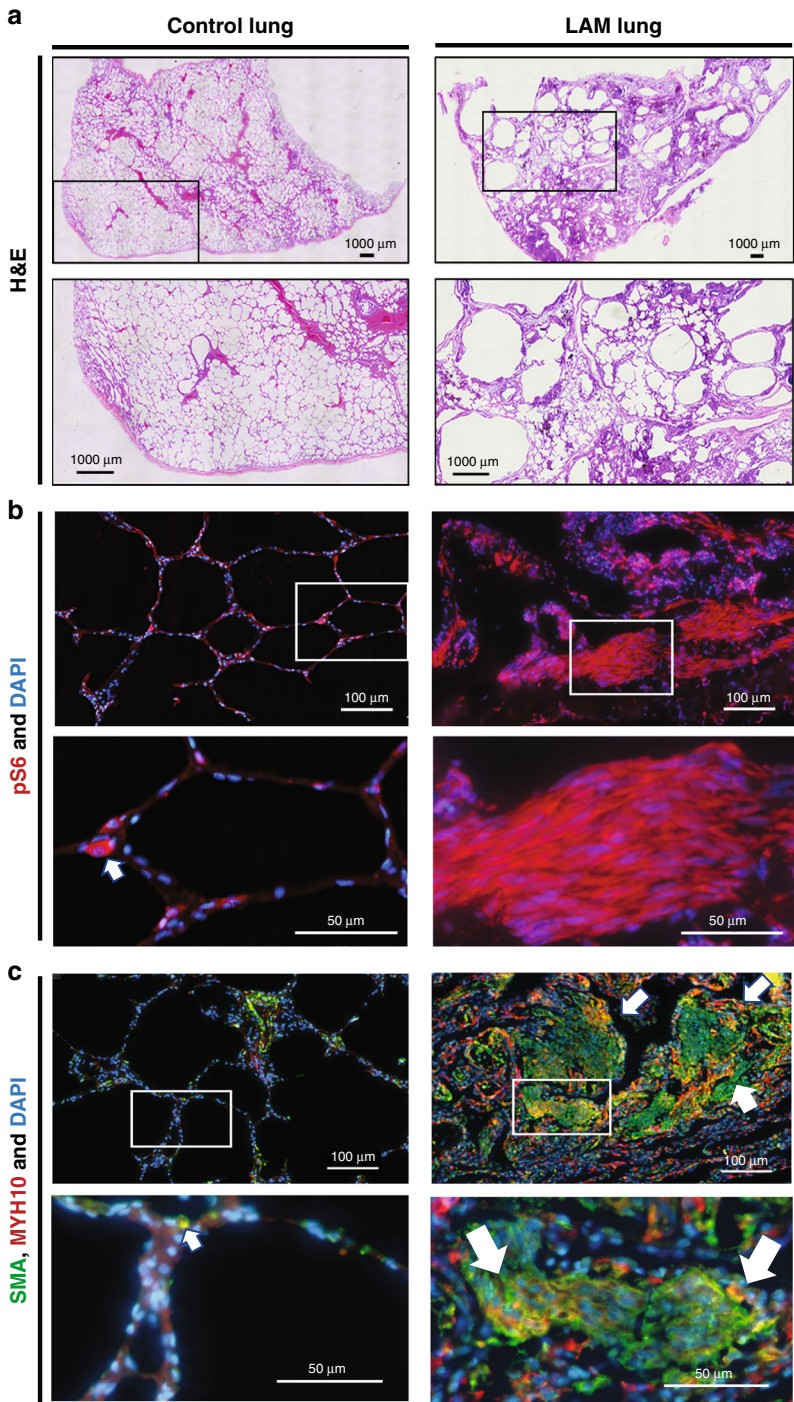

**Fig. 1 Morphology and histopathology of LAM lung. a** Images of the H&E staining of the control human (on the left) and LAM lung (on the right) sections**. b** mTORC1 activation detected with anti-phospho-ribosomal protein S6 (pS6) antibody (red) in a single cell of the control lung (shown with arrowhead) and the entire LAM lung lesion. **c** Immunofluorescent staining of the human lung sections with (SMA, green), anti-Myosin Heavy Chain 10 (MYH10) (red) antibodies, and DAPI (blue) for the nuclei. Arrows point out the colocalization of SMA and MYH10 in a single cell in control lung and LAM lesions.

an equivalent in a healthy control lung. It was characterized by high differential expression of mesenchymal (*PRG4, OGN, MYH10*), endothelial (*VCAM1, PDPN*), and epithelial (*KRT18, MSLN*) gene markers (Fig. 2c, d, Supplementary Fig. 3A, Supplementary Table 1), which made it challenging to assign it to a specific cell type. To define the relation of the LAM lung cells to the other cells in the LAM lung, we compared the top 100 differentially expressed genes in LAM lung cells to the rest of the cell clusters and found that they were the most transcriptional

proximal to the MACs subset (Supplementary Fig. 3A), suggesting a likely mesenchymal origin. Interestingly, among the genes expressed both in LAM lung cells and MACs, we found a number of genes that belonged to the estrogen receptor *alpha* (ERα) transcriptional targets[22,23] (Supplementary Fig. 3A). Notably, LAM MACs showed increased expression of the *ESR1* gene, encoding the ERα receptor, compared to that of control MACs, suggesting the involvement of estrogen regulation in modulating LAM MACS transcription (Supplementary Fig. 3B, C). Compared

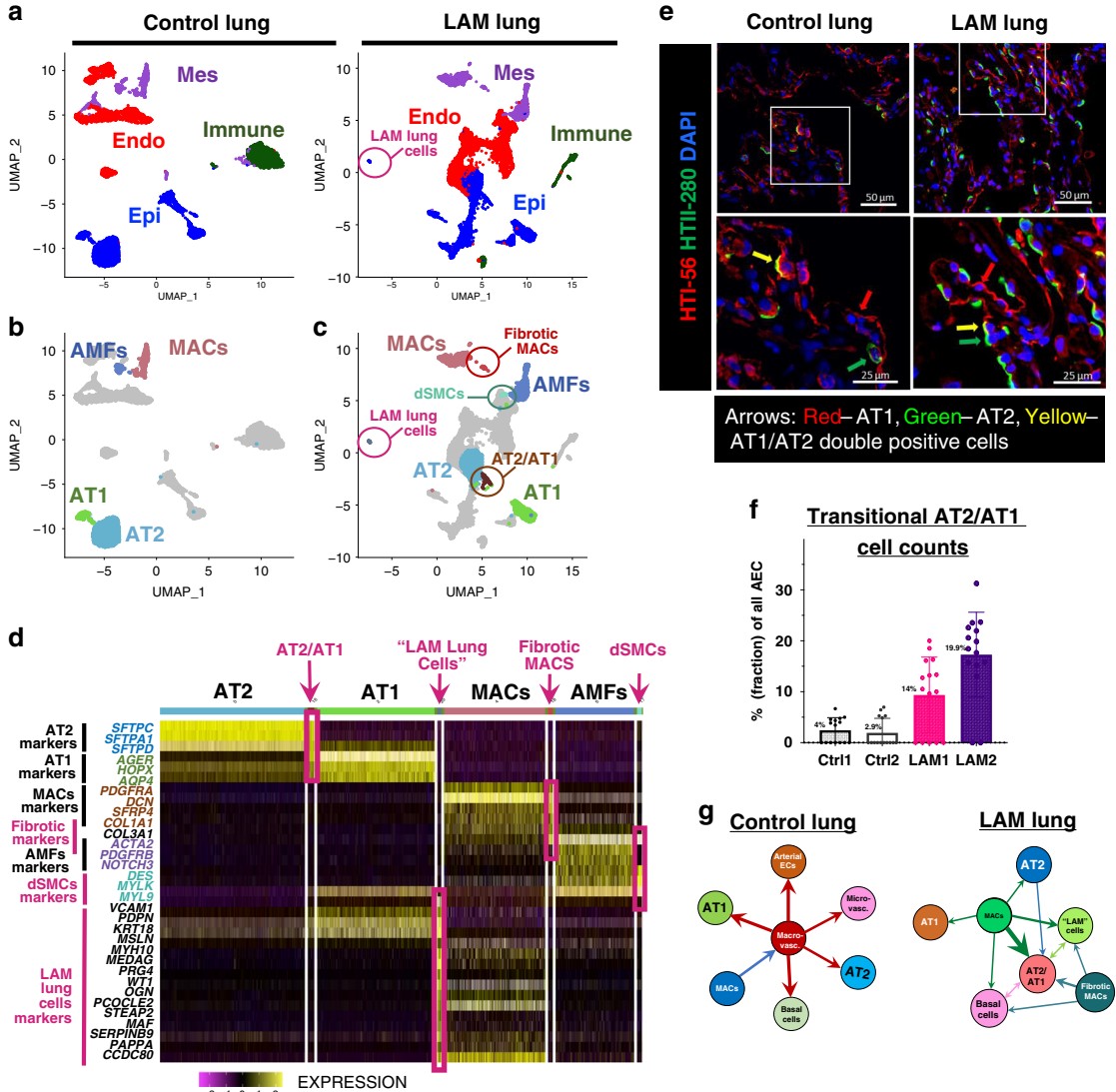

**Fig. 2 Single-cell RNA-seq identifies distinct LAM lung cells and transitional alveolar epithelial and mesenchymal cells. a** UMAP dimensionality reduction plots represent scRNA-seq analysis of the lungs from the LAM patient ($N = 2$) and age- and sex-matched control donor ($N = 2$). Colored by the core cellular niches are epithelial (blue), endothelial (red), mesenchymal (purple), and immune cells (green). **b** UMAP of control human lung highlighting AT2 cells (light blue), AT1 cells (light green), mesenchymal alveolar cells (MACs) (pink), and alveolar myofibroblasts (AMFs) (blue). **c** UMAP highlighting LAM lung epithelial cell clusters including: AT2 cells (light blue), AT1 cells (light green), and transitional AT2/AT1 cell subtype (brown); mesenchymal cell clusters including: MACs (pink), fibrotically activated MACs subtype (red), AMFs (blue), differentiated smooth-muscle cells (dSMCs) (cyan); and unique LAM lung cells cluster (purple). **d** Differential gene expression of the top differentially expressed gene markers for described cell types (AT2, AT1, MACs, AMFs) and the distinct LAM lung cells (highlighted in red, indicated with arrows). Heatmap illustrates the shared expression of marker genes of the canonical cell types in the transitional cell states and the distinctive LAM lung-specific cell types. **e** Immunofluorescent staining of human control lungs and LAM lung sections for the specific AT1 and AT2 cell markers (HTI-56 and HTII-280, respectively). Representative confocal images of alveolar areas. **f** Quantitation of the AT2/AT1 transitional state cell fraction in the total number of AT1 and AT2 cells per image area. Based on the experiment shown in the panel **e**, performed on $N = 2$ for control, and $N = 2$ for LAM lungs, with $N = 10$ of independent confocal images analyzed in each sample. Error bars represent mean values with SD. Raw data and statistical details are supplied in the Source data file. **g** Models of major cellular crosstalk directions in LAM and control lungs based on gene expression of the ligand-receptor pairs. The arrows show the direction of the crosstalk and the thickness of the arrow indicates the number of interactions.

to the control equivalent, LAM MACs also exhibited increased expression of *VEGFD*, a known biomarker of LAM (Supplementary Fig. 3B, C). Among the LAM lung cells gene candidates potentially regulated by estrogen, we highlighted the expression of *MYH10*, encoding myosin heavy chain 10 protein, or non-muscle myosin IIB, which was previously shown to be associated with LAM[24]. To validate our findings, we demonstrated the upregulation of the *MYH10* gene in LAM lungs (Supplementary Fig. 3C) and marked expression of MYH10 protein in LAM lung lesions of

LAM patient lung tissue specimens using fluorescent immunohistochemistry (Fig. 1c).

To further investigate the link between MACs and the LAM lung cells cluster, we analyzed putative cellular crosstalk directions between all cell types both in LAM and control lungs based on the scRNA-seq expression of the known ligand-receptor pairs[25] (Fig. 2g). Confirming our hypothesis, we have found that in contrast to the control lung, MACs appeared in the center of the crosstalk in LAM lung, while specifically enriched interactions

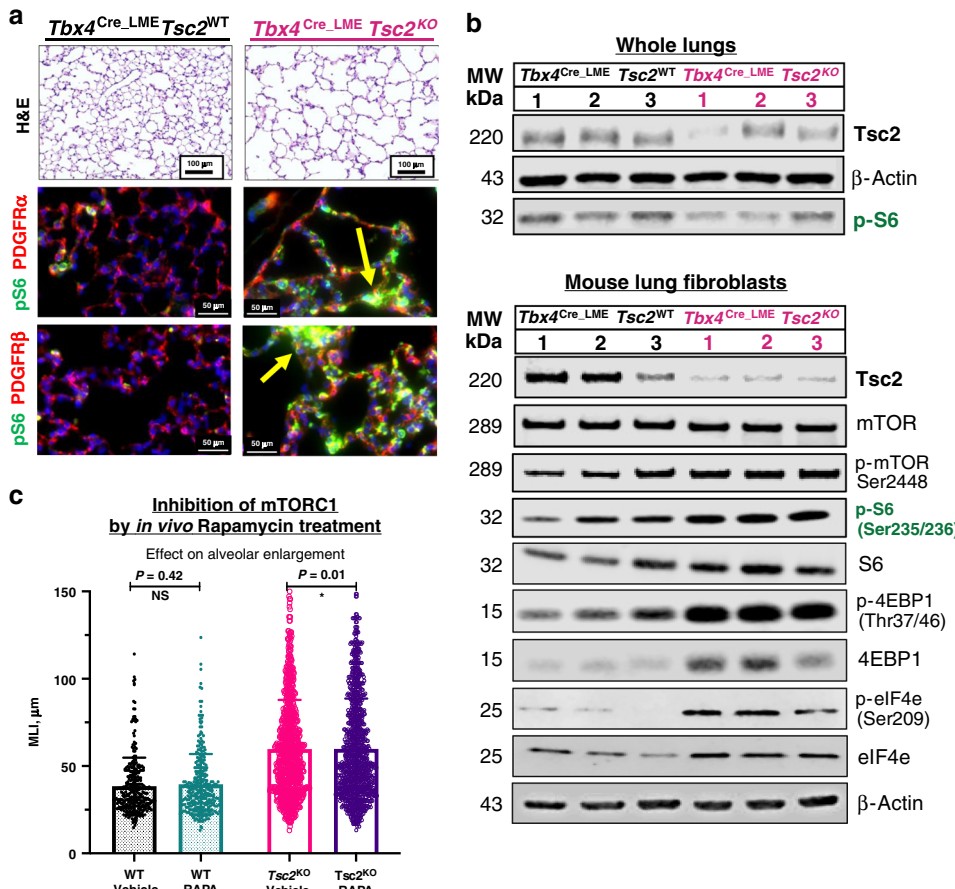

**Fig. 3 Tsc2 KO in mouse lung mesenchyme leads to mTORC1 pathway activation and alveolar enlargement. a** Alveolar enlargement in $Tbx4^{LME\_Cre}Tsc2^{KO}$, visualized by H&E staining of the 8-week-old mouse lung sections. Tsc2-dependent mTORC1 activation in lung mesenchyme in vivo visualized by dual immunofluorescent staining of the consecutive sections of the same mouse lungs for pS6 (green) and the markers of the two major mesenchymal subsets: PDGFRα, PDGFRβ (red) ($N = 6$, three independent experiments). **b** Immunoblots of the whole lung lysates from 12-week-old $Tbx4^{LME\_Cre}Tsc2^{WT}$ (WT) and $Tbx4^{LME\_Cre}Tsc2^{KO}$ (KO) mice and mouse lung fibroblasts (MLFs) derived from those lungs, showing TSC2 protein loss and expression of molecular markers of mTORC1 pathway activation ($N = 3$). **c** Inhibition of mTORC1 by Rapamycin (administered in vivo for 4 weeks) slows down alveolar enlargement of the 8-week-old $Tbx4^{LME\_Cre}Tsc2^{KO}$ mice. Mean linear intercept (MLI) used as a measure of the alveolar size ($N = 6$ per group, $N = 15$ images per lung). Error bars represent mean values with SD. $P$ values were obtained from multiple $T$-test group comparison. Statistical significance determined using the Holm–Sidak method, with alpha = 0.05. Each row was analyzed individually, without assuming a consistent SD. Raw data supporting the figure panels **b** and **c** are included in the Source data file.

were pointing towards both AEC cell types, transitional AT2/AT1 cells, as well as LAM lung cells.

Collectively, this data demonstrates that many of the pathological changes in LAM lung may have been induced by the abnormal signaling originating from mesenchymal cells, specifically from LAM cells and MACs. We propose that the altered LAM and MACs signaling may have triggered the formation of transitional AT2/AT1 cell state, and the complex cellular interplay may have orchestrated the pathological lung tissue remodeling observed in LAM lungs.

**Lung-mesenchymal Tsc2<sup>KO</sup> leads to mTORC1 activation and the alveolar enlargement**. In the scRNA-seq analysis of LAM lung, we found the unique LAM lung cells exhibiting close transcriptional similarity to MACs (Fig. 2d and Supplementary Fig. 3A) which suggested their potential mesenchymal origin. Based on this data we proposed that loss of TSC2 in the lung-mesenchymal progenitors may induce a mouse lung phenotype similar to human LAM. Thus, we created a conditional knock-out of the Tsc2 gene specifically in lung mesenchyme by crossing $Tsc2^{f/f}$ mice with $Tbx4^{LME\_Cre}$ mice[26],

in which a lung-mesenchymal enhancer element from Tbx4 locus was combined with an Hsp68 minimal promoter to drive Cre expression exclusively in the lung mesenchyme. While $Tbx4^{LME\_Cre}Tsc2^{KO}$ mice were viable, healthy, and fertile, the gross lung morphology revealed significantly enlarged alveolar spaces compared to WT controls (Fig. 3a). To validate the specificity of the Tsc2 KO in vivo we co-stained consecutive lung sections of the 8-week-old $Tbx4^{LME\_Cre}Tsc2^{KO}$ mouse lungs for the pS6 and the markers of the two major mesenchymal subsets: PDGFRα and PDGFRβ. We detected mTORC1 activation in both of these subpopulations originating from the lung mesenchyme progenitor niche (Fig. 3a). We validated the Tsc2 gene KO by the decrease in TSC2 protein and consecutive mTORC1 pathway upregulation in whole lung lysates and mouse lung fibroblasts (MLFs) isolated from $Tbx4^{LME\_Cre}Tsc2^{KO}$ mouse lungs (Fig. 3b), with a subsequent increase in pS6, p4E-BP1, and peIF4E, downstream molecular targets of mTORC1 activation (Fig. 3b). To test whether inhibiting the activated mTORC1 will have a protective effect on the development of the alveolar enlargement in $Tbx4^{LME\_Cre}Tsc2^{KO}$ mouse lungs, we administered rapamycin (allosteric inhibitor of mTORC1) three times a week, via i.p. injections for 4 weeks. Subsequent analysis of

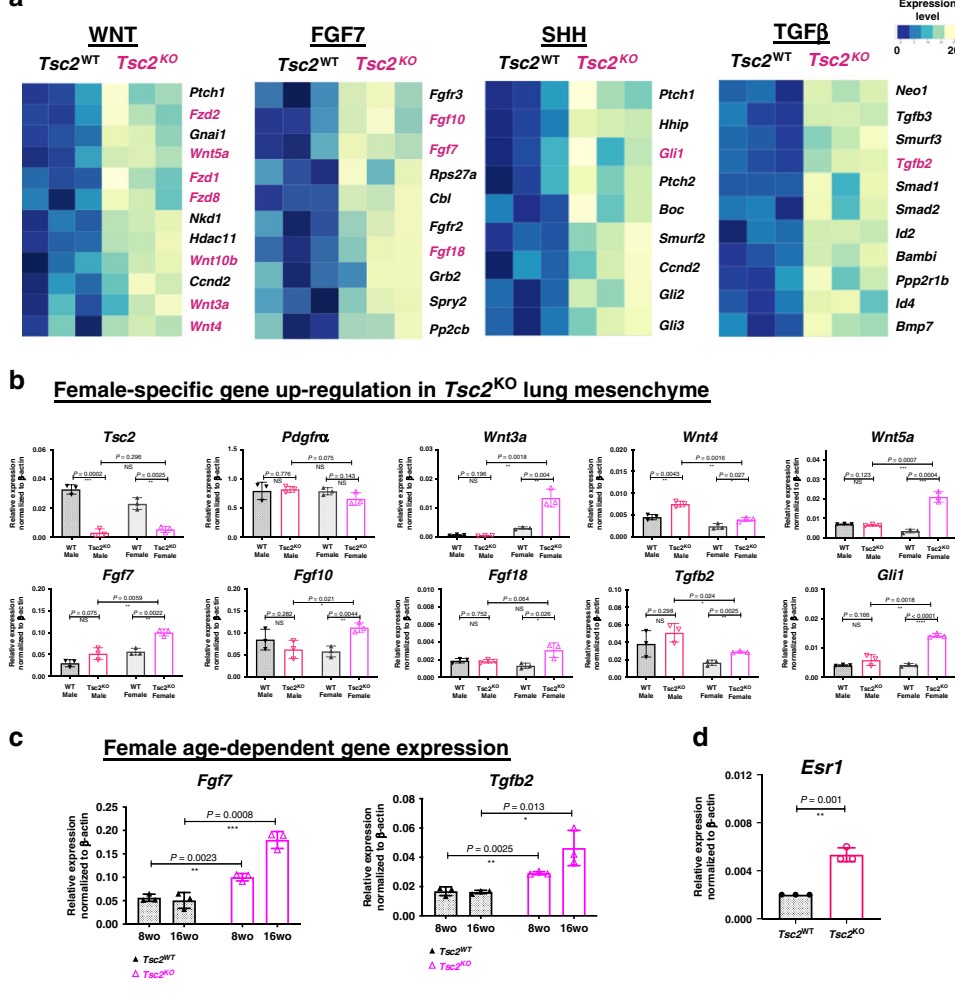

**Fig. 4 Sex- and age-specific gene expression changes in the *Tbx4*^LME_Cre*Tsc2*^KO lung mesenchyme. a** FGF7, TGFB, and SHH pathways upregulation in *Tsc2*^KO mouse lung mesenchyme. PopRNA-seq data, N = 3. Highlighted genes were selected as targets for qRT-PCR validation. **b** Validation of the popRNA-seq targets in vivo. Bar graphs represent qRT-PCR data for the gene expression in the lung-mesenchymal cells. Analysis was performed on the mRNA samples isolated from the lung mesenchyme of a separate set of the 8-week-old *Tbx4*^LME_Cre*Tsc2*^WT and *Tbx4*^LME_Cre*Tsc2*^KO mouse lungs, males and females, N = 3 each, using magnetic sort as depicted in the Supplementary Fig. 4a. The data points represent relative gene expression values normalized to the expression of β-actin gene using ddCt method. **c** Selected examples of the age-dependent increase in the gene expression of *Fgf7* and *Tgfb2* genes in mesenchymal cells isolated from the 8- and 16-week-old female *Tbx4*^LME_Cre*Tsc2*^KO and *Tbx4*^LME_Cre*Tsc2*^WT mouse lungs. **d** Increased expression of the *Esr1* gene (encoding Estrogen Receptor α) specifically in the *Tsc2*^KO MLFs isolated by the magnetic sort from the 8-week-old female WT and *Tbx4*^LME_Cre*Tsc2*^KO lungs. All bar graphs represent mean values +/− SD. N = 3 for each experimental group. P values, means, and SD error bars were obtained by multiple non-parametric *T*-test comparisons. Raw data underlying graphs in panels **b**–**d** are supplemented in a Source data file.

lung morphology of the 8-week-old mouse lung sections showed a mild but statistically significant protective effect of rapamycin in the experimental group of *Tbx4*^LME_Cre*Tsc2*^KO mice compared to the WT and vehicle controls (Fig. 3c).

**WNT inhibition in *Tsc2*^KO mesenchyme protects mouse lungs from alveolar enlargement in an age-dependent manner.** To elucidate molecular mechanisms underlying the observed phenotype, we examined transcriptomic changes induced by *Tsc2* KO in the lung mesenchyme of the 8-week-old female *Tbx4*^LME_Cre*Tsc2*^KO lungs compared to sex- and age-matched WT controls. Bulk RNA-seq analysis of the sorted major lung cell populations (Pop-seq), including immune (CD45^+), epithelial (EpCAM^+), endothelial (CD31^+, PDPN^+), and mesenchymal (all negative) cells was performed using consecutive depletion of cell fractions (Supplementary Fig. 4a). Enrichment of the gene expression for the cell-specific markers of the sorted cells was confirmed by qPCR (Supplementary Fig. 4b).

As expected, *Tsc2*^KO mesenchymal cells presented global gene expression changes, compared to the WT control, including the downregulation of the genes involved in oxidative phosphorylation, fatty acid metabolism pathways, and the upregulation of the hypoxia pathway, which were known to be associated with dysregulated mTORC1 signaling[6] (Supplementary Fig. 4c). However, we additionally found increased expression of the genes involved in the WNT, FGF, TGFB, and SHH pathways (Fig. 4a). We validated the upregulation of the select genes in these pathways by qPCR on a separate set of animals and noticed that some of these genes were upregulated only in female *Tbx4*^LME_Cre*Tsc2*^KO mice (Fig. 4b–d). However, we specifically focused on *WNT*-ligand encoding genes, since the WNT pathway is well known to be critical in lung development, maintenance of lung homeostasis, lung regeneration[27–29], and in lung diseases[30]. We found that several genes related to the WNT pathway, including *Wnt3a, Wnt4, Wnt5a*, and *Wnt10b* ligands, *Fzd1* and *Fzd8* receptors, and *Hdac11* and *Ccnd2* WNT-response genes were upregulated in

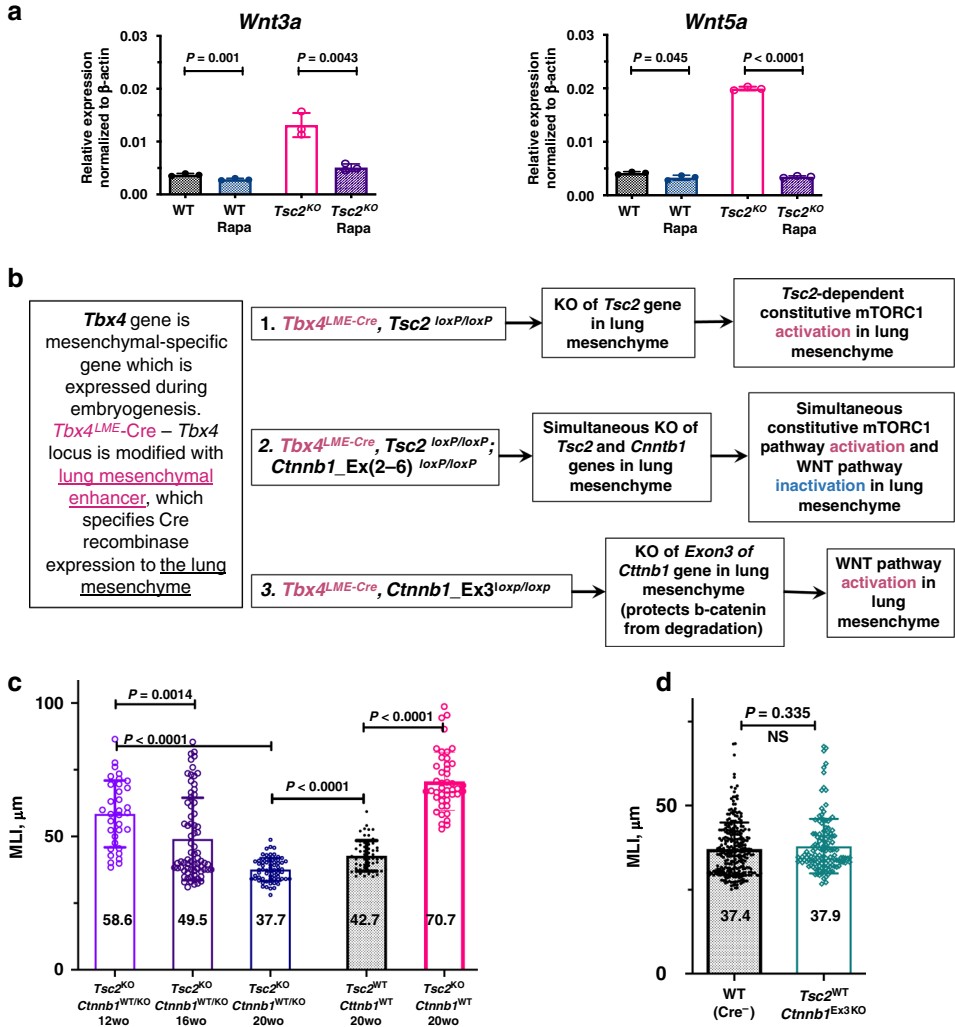

**Fig. 5 WNT pathway inhibition in *Tsc2* KO mouse lung mesenchyme protects from alveolar enlargement. a** Wnt5a and Wnt3a genes are downregulated by in vivo Rapamycin (Rapa) treatment in the 8-week-old *Tsc2*KO lung mesenchyme. qRT-PCR results on the sorted MLFs after 4 weeks of Rapamycin injections ($N = 3$ per group). **b** Detailed diagram of the mouse genotypes used in the study for the lung-mesenchymal-specific gene knock-outs and expression. **c** Inhibition of the WNT pathway in vivo by the β-catenin KO in *Tsc2*KO mouse lung mesenchyme decreases the size of alveoli with age ($N = 5$, 15 images per lung). **d** WNT pathway activation by stabilizing β-catenin in *Tsc2*WT mesenchyme does not trigger the alveolar enlargement. Plot shows MLIs of 20-week-old mouse lungs ($N = 5$, 15 images per lung). For all charts in this figure, error bars represent mean values + SD. *P* values obtained by multiple non-parametric two-tailed *T*-tests in a group comparison. Statistical significance determined using the Holm–Sidak method, with alpha = 0.05. Each row was analyzed individually, without assuming a consistent SD. Raw data underlying the graphs in the panels **a**, **c**, and **d** is supplied in the Source data file.

*Tsc2*KO lung mesenchyme (Fig. 4a). Notably, *Wnt3a* and *Wnt5a* ligand encoding genes appeared sensitive to rapamycin treatment in vivo (Fig. 5a), suggesting the mTORC1-dependent mechanism of their increased transcription.

To further explore the role of WNT pathway in the *Tbx4*LME_Cre *Tsc2*KO lung phenotype, we crossed *Tbx4*LME_Cre*Tsc2*f/f and *Ctnnb1*(exon2-6)f/f mice[31] creating a double conditional KO of *Tsc2* and *Ctnnb1* (encoding β-catenin) genes in lung-mesenchymal cells— *Tbx4*LME_Cre*Tsc2*KO*Ctnnb1*KO mice (Fig. 5b). Subsequent analysis of the lung phenotype in these mice revealed that the addition of WNT pathway inhibition to the *Tsc2*-dependent mTORC1 activation in the lung mesenchyme exerts a significant age-dependent protective effect, as evident from gradual MLI decrease in 12-, 16-, and 20-week-old mice (Fig. 5c). In fact, at the age of 20 weeks, the size of the alveoli was restored to the WT control levels. This observation underscored the important role of the molecular interplay of the WNT and TSC2/mTORC1 pathways in the development of the alveolar enlargement phenotype in *Tbx4*LME_Cre*Tsc2*KO mouse lungs. The activation of the WNT pathway alone in the lung mesenchyme through stabilization

of β-catenin in *Tbx4*LME_Cre*Ctnnb1*Ex3KO mice, however, did not trigger the alveolar enlargement (Fig. 5d).

**Tsc2KO mesenchyme affects alveolar epithelial cell fitness.** Despite the deletion of *Tsc2* only in mesenchymal cells (Fig. 6b), epithelial cells in *Tbx4*LME_Cre*Tsc2*KO lungs also experienced significant global gene expression changes (Supplementary Fig. 4C). Specifically, we noted the upregulation of WNT-response genes *Axin1*, *Axin2*, *Fzd1*, and *Fzd2* (Fig. 6a) which could be triggered by the upregulated WNT ligands produced by the surrounding *Tsc2*KO mesenchymal cells (Fig. 4a). It has been previously shown, that the subsets of lung-mesenchymal cells expressing WNT ligands activate WNT signaling in Axin2+ AT2 cells, which serve as alveolar progenitor cells for AT1 and AT2 cells to enhance alveolar repair[27,32]. Interestingly, similar to that of observed in the LAM lung, we have found the increased transition of AT2 to AT1 cells in *Tbx4*LME_Cre*Tsc2*KO lungs in vivo (Fig. 6c), while the shift in AT2/AT1 ratio was observed specifically in females and progressed with age (Fig. 6d). To test

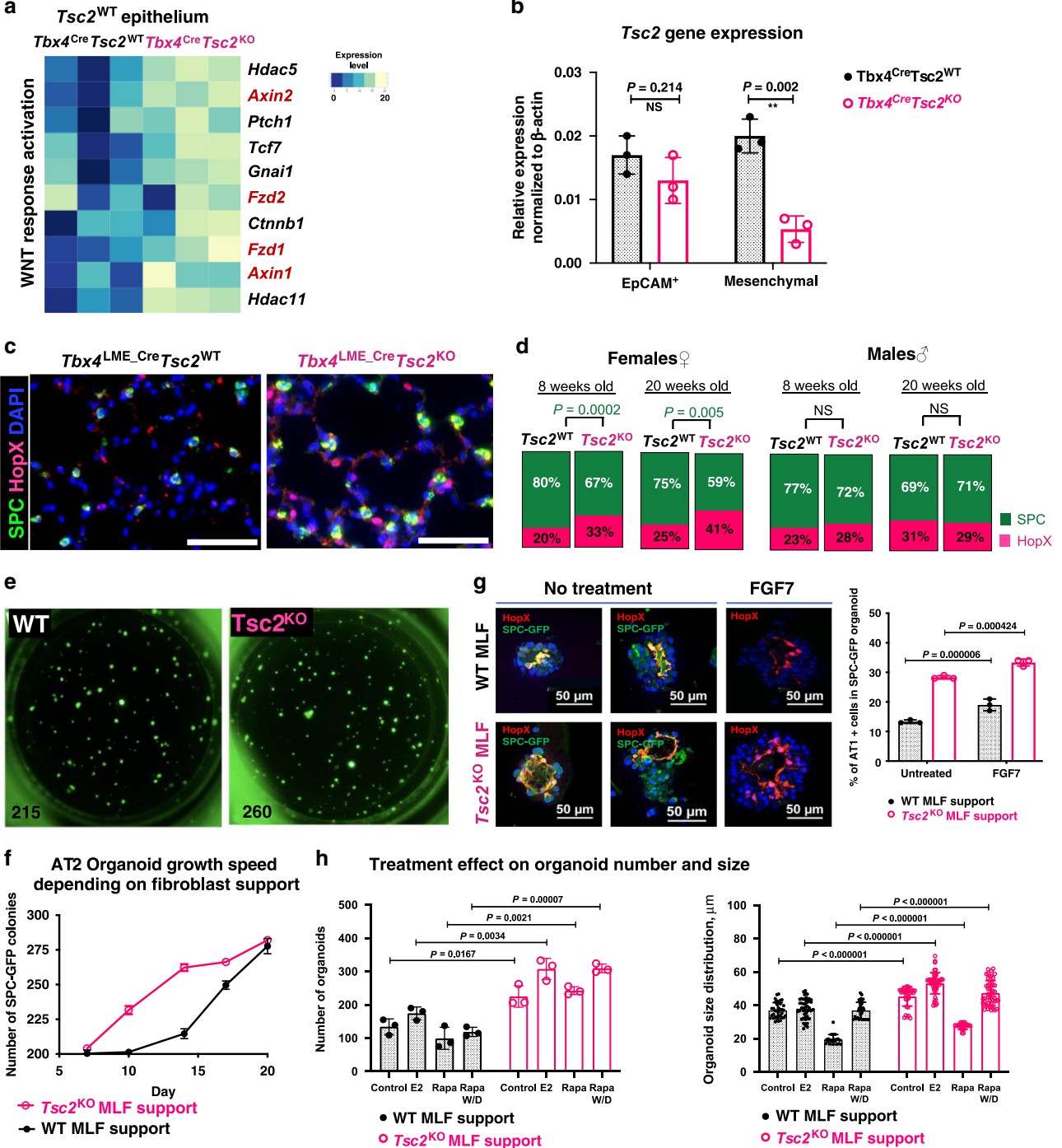

**Fig. 6 Tsc2 KO MLFs alter AT2 cells fitness in vivo and ex vivo. a** Upregulation of WNT-response genes in the $Tbx4^{Cre}Tsc2^{KO}$ lung epithelial cells (Pop-seq results, 8-wo mice, $N = 3$). **b** Tsc2 gene expression in $Tsc2^{KO}$ lung mesenchyme and $Tsc2^{WT}$ lung epithelium ($N = 3$). **c** Representative images of the 8-week-old female $Tbx4^{LME\_Cre}Tsc2^{WT}$ and $Tbx4^{LME\_Cre}Tsc2^{KO}$ lungs stained for AT2 and AT1 cell markers: SPC (green) and HopX (pink), respectively. **d** Quantitative analysis of AT2/AT1 ratio in the 8- and 20-wo lungs from female and male mice based on staining represented in panel **c** ($N = 3$, 6 independent images per lung). Graphs show % of each cell type (AT2 or AT1) in total number of AECs. $P$ values obtained for the non-parametrical paired $t$-test). **e** Representative images of AT2 organoid (GFP$^+$) colony-forming efficiency (CFE) co-cultured with $Tsc2^{KO}$ or $Tsc2^{WT}$ MLFs. **f** Changes in the CFE over time based on experiment represented in panel **e** (error bars show means $+/-$ SD, $N = 3$ for each time point). **g** $Tsc2^{KO}$ MLFs stimulate the AT2→AT1 transition within the organoid colonies. AT2+ and AT1+ cells counts are based on SPC (green) and HopX (red) staining of the embedded and fixed organoid sections taken at the terminal time point—21 days. The graph shows the percentage of AT1 cells from the total DAPI$^+$ cells taken as 100%. Error bars indicate SD for the non-parametric $t$-test for the $N = 3$ bio repeats. **h** $Tsc2^{KO}$ MLFs stimulate CFE of the AT2 organoids, compared to $Tsc2^{WT}$ MLFs; Estrogen (E2) stimulates CFE and size of AT2 organoids, specifically with $Tsc2^{KO}$ MLF support; Rapamycin (Rapa) inhibits CFE and reduces their size. Data collected at the terminal time point 21 days. Error bars reflect means $+/-$ SD for $N = 3$ bio replicates in each condition. $P$ values in all group comparisons obtained from multiple two-tailed $T$-tests. Statistical significance determined using the Holm–Sidak method, with alpha = 0.05. Each row was analyzed individually, without assuming a consistent SD. Raw data used for plotting the graphs in panels **b**, **d**, **g**, **f**, and **h** is supplied in a Source data file.

the hypothesis that the $Tsc2^{KO}$ lung mesenchyme may impact the AT2 cell fitness, we utilized ex vivo 3D alveolosphere assay. In our model systems, cultured in Matrigel for 21 days, $Tsc2^{KO}$ MLF support cells accelerated growth and increased the colony-forming efficiency (CFE) of the AT2 organoids (Fig. 6e, f), as well as enhanced the transdifferentiation from AT2 to AT1 cells (Fig. 6g), compared to $Tsc2^{WT}$ support cells. Interestingly, estrogen treatment increased the number and size of AT2 organoids specifically with $Tsc2^{KO}$ MLF support, and rapamycin had a partial inhibitory effect (Fig. 6h). However, following rapamycin withdrawal, organoid colonies reached the original control mean size (Fig. 6h), which correlates with the known cytostatic effect of rapamycin.

**Mesenchymal $Tsc2^{KO}$ leads to the age-dependent female-specific lung function decline.** We have discovered that the increase in alveolar size in $Tbx4^{LME\_Cre}Tsc2^{KO}$ mouse lungs progressed with age, as evident from MLI changes at the age of 8-, 12-, and 20-weeks old (Fig. 7a). To evaluate the physiological relevance of the observed increased airspaces, we performed pulmonary function tests (PFTs) on $Tbx4^{LME\_Cre}Tsc2^{KO}$ and $Tbx4^{LME\_Cre}Tsc2^{WT}$ control mice (Fig. 7b, c). At 12- and 24 weeks of age, the inspiratory capacity of male $Tbx4^{LME\_Cre}Tsc2^{KO}$ mice was comparable to the WT controls. An increase in compliance and decrease in elastance was observed in 12-week-old male $Tbx4^{LME\_Cre}Tsc2^{KO}$ mice, but the lung function stabilized and was comparable to the WT at 24 weeks (Fig. 7c). The PFTs of 12-week-old female $Tbx4^{LME\_Cre}Tsc2^{KO}$ mice were comparable to WT controls. However, by 24 weeks, they developed a marked and statistically significant decline in pulmonary function: significantly increased lung inspiratory capacity (IC), combined with increased lung compliance ($C_{RS}$) and decreased elastance ($E_{RS}$) (Fig. 7c).

Pregnancies exacerbate $Tbx4^{LME\_Cre}Tsc2^{KO}$ lung structure decline. When analyzed the temporal changes in the $Tbx4^{LME\_Cre}Tsc2^{KO}$ mouse lung phenotype, we found the most pronounced age-dependent loss of alveolar structure in female mice at the age of 54 weeks (Fig. 8a). However, these changes did not correlate with the loss of alveolar septal thickness (Fig. 8b), suggesting that the decrease in extracellular matrix proteins to be the unlikely primary cause.

In LAM disease, the worsening of the lung function is associated with estrogen concentration during the menstrual cycle, and the disease is exacerbated by pregnancy[5]. Previously we have noticed that $Esr1$ gene expression was increased in $Tbx4^{LME\_Cre}Tsc2^{KO}$ mesenchyme (Fig. 4d). We determined the effect of multiple pregnancies on the lung structure of 54-week-old female $Tbx4^{LME\_Cre}Tsc2^{KO}$ mice compared to WT controls (Fig. 8c). $Tbx4^{LME\_Cre}Tsc2^{KO}$ female mice, who had undergone pregnancies (breeders) showed significantly increased alveolar sizes compared to the non-breeder $Tbx4^{LME\_Cre}Tsc2^{KO}$ female mice (Fig. 8a). Breeder $Tbx4^{LME\_Cre}Tsc2^{KO}$ mice also developed microscopic and distal lung lesions compared to the non-breeder $Tbx4^{LME\_Cre}Tsc2^{KO}$ mice or WT breeders and non-breeders (Fig. 8c, d). These lesions stained positive for pS6, SMA, and MYH10 (Fig. 8e), which correlates with the activation of mTORC1 signaling, and the upregulation of SMA and MYH10 in human LAM lesions (Fig. 1c). Our data demonstrate that pregnancies exacerbate lung structure decline in the mouse lung-mesenchymal $Tsc2$ KO model which correlates with the deleterious effect of pregnancies in LAM disease.

## Discussion

The constitutive activation of mTORC1 in LAM lung causes cystic airspace enlargement, spontaneous lung collapse, and respiratory failure. The specific cell types driving these pathological changes in the lung have not been well-defined. Here, we used single-cell profiling of a human LAM lung compared to an age- and sex-matched control lung as a hypothesis-generating approach. The scRNA-seq analysis revealed distinct transitional states of mesenchymal (fibrotically-activated MACs) and alveolar epithelial cells (AT2/AT1) in the LAM lung, as well as identified a cellular subtype unique for the LAM lung–LAM lung cells. Notably, LAM MACs were similar to the fibroblasts identified in pulmonary fibrosis, which also express $COL3A1$, $COL1A1$, and $ACTA2$[33]. MACs from LAM lung also expressed $SPINT2$, $FGFR4$, $ITGA8$, $SFRP2$, and $SERPINF1$ genes identified in mesenchymal cells from human lung stroma[34].

To verify the transitional state of AECs, observed in the scRNA-seq, we discovered a consistently increased number of transitional AT2 cells expressing AT1 cell markers in the lung parenchyma of LAM lungs and a significantly altered architecture of the alveoli septa. In the adult lung, a sub-lineage within the AT2 cell population has been shown to play a role in the repair of the lung alveoli[27,32]. A new transitional state of AT2/AT1 cells identified in the LAM lung demonstrates the dysregulation of their quiescent state and suggests a state of potentially active chronic alveolar repair, which may have a deleterious effect on the regenerative capacity of AECs in the LAM lung. Interestingly, recently published studies also mentioned alveolar epithelial cell plasticity in relation to regeneration after injury[35,36]. We have compared our AT2/AT1 transitional cell subtype from the LAM lung to the "pre-alveolar-type-1 transitional cell state" (PATS)[36] and found similarities in their transcriptional signatures. Specifically, the PATS gene markers $LGALS$, $SFN$, $CLDN4$, and $KRT8$ were also expressed in the transitional AT2/AT1 cell subtype discovered in the LAM lung, which further links them to the state of regeneration.

The most distinctive cell type found in the scRNA-seq of the LAM lung was LAM lung cells, which had no equivalent in the control lung. Due to the limitations of the scRNA-seq method we were unable to detect point mutations or to show the loss of $TSC1$ or $TSC2$ gene expression in the LAM lung cell subset, nor the other cell types in the LAM lung. However, when compared to the differentially expressed genes in LAM lung cells and the upregulated genes in the known TSC2-null human LAM cell line – kidney angiomyolipoma (AML) cells[37,38], we found a number of genes in common (Supplementary Fig. 3D). Many of these genes appeared to be related to the mTORC1 pathway activation. Further evaluation of the LAM lung cell cluster revealed their closest transcriptional proximity to MACs. Additional evidence of the link between LAM lung cells and MACs was a number of ERα genes that both cell types had in common. Furthermore, LAM MACs in contrast to the control expressed higher levels of $ESR1$ gene, suggesting that under conditions of high estrogen, MACs could have undergone a significant transformation due to the estrogen-regulated transcriptional changes. Together these findings pointed us to a hypothesis that loss of $Tsc2$ in mouse lung mesenchyme could lead to the development of a LAM-like phenotype.

Indeed, our mouse model of mTORC1 activation by selective $Tsc2$ deletion in lung progenitor mesenchymal cells demonstrated the age- and sex-linked structural and functional decline of the lung and resembled many of the characteristics of human disease LAM. We have to note, that in contrast to the observed mouse lung morphology, which shows a relatively uniform peripheral enlargement of the airspaces, LAM lung has localized enlarged cystic airspaces (Fig. 1a). This difference may be explained by that in our model TSC2 was deleted uniformly throughout the mouse lung parenchyma, while in human LAM lung TSC2 loss occurs only in a small subset of cells. However, like in human LAM, we have discovered the exacerbating effect of pregnancies on the lung structure decline and the lesion formation.

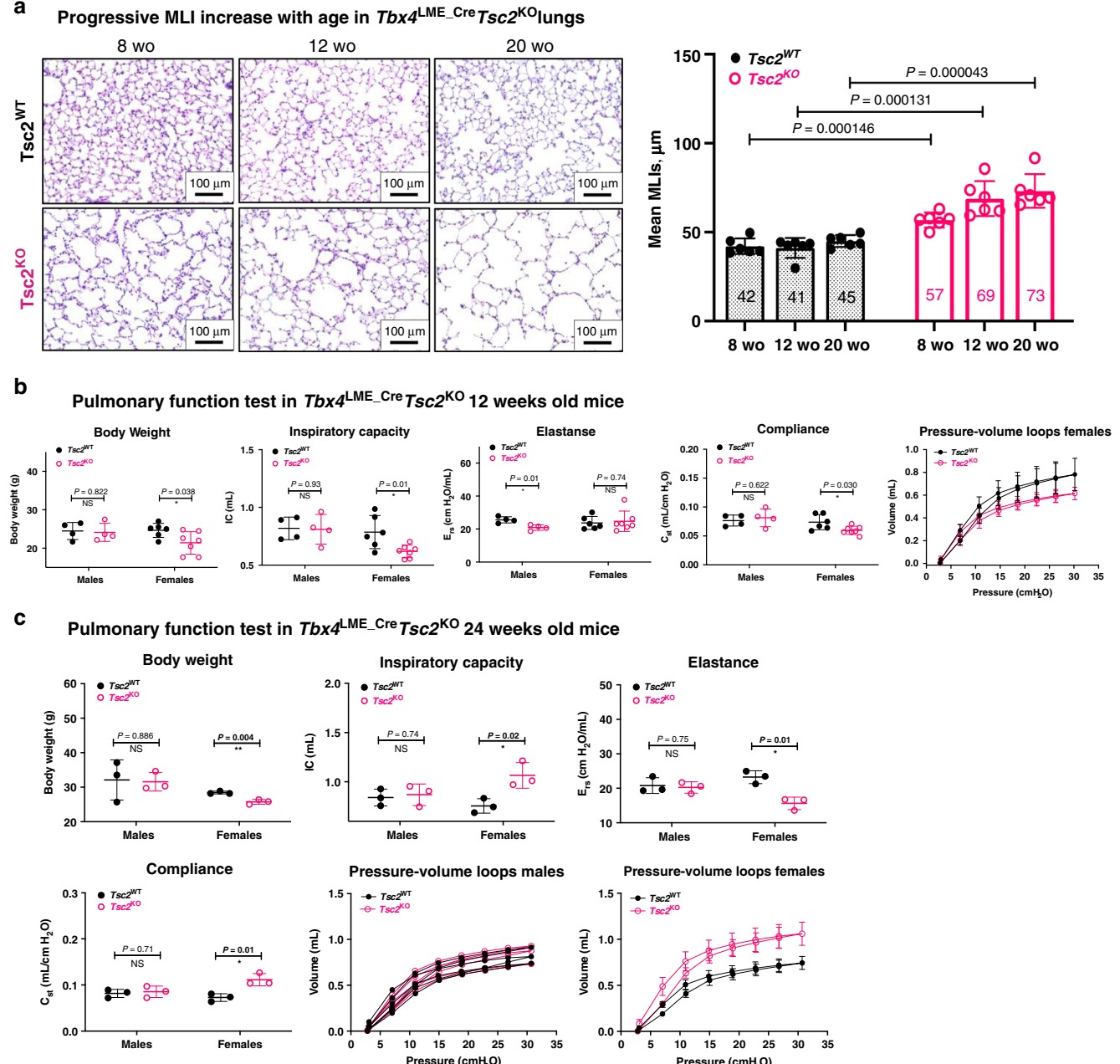

**Fig. 7 Age- and sex-dependent lung structure and function decline in *Tbx4*<sup>LME_Cre</sup>*Tsc2*<sup>KO</sup> mice. a** Representative images of the H&E staining of the lung sections of 8-, 12-, and 20-week-old *Tbx4*<sup>LME_Cre</sup>*Tsc2*<sup>WT</sup> and *Tbx4*<sup>LME_Cre</sup>*Tsc2*<sup>KO</sup> mice. Experiments were reproducible which is evident from the adjacent quantitative assessment using high-throughput MLI analysis. *N* = 6 in each group. **b** Pulmonary function test in the 12-week-old *Tbx4*<sup>LME_Cre</sup>*Tsc2*<sup>WT</sup> and *Tbx4*<sup>LME_Cre</sup>*Tsc2*<sup>KO</sup> mice performed by flexiVent. Experimental groups were divided by sex with *N* = 4 for WT, *N* = 6 for KO animals per group. Pressure-volume (PV) loops are displaying only the individual deflation limbs for clarity. **c** FlexiVent parameters for the 24-week-old *Tbx4*<sup>LME_Cre</sup>*Tsc2*<sup>WT</sup> and *Tbx4*<sup>LME_Cre</sup>*Tsc2*<sup>KO</sup> mice. *N* = 3 for each group. The results in all graphs show mean values ± SD. *P* values obtained from multiple *T*-test group comparison. Statistical significance determined using the Holm–Sidak method, with alpha = 0.05. Each row was analyzed individually, without assuming a consistent SD. Raw data underlying all graphs is supplied in a Source data file.

As expected, *TSC2* KO in lung mesenchyme triggered a subsequent constitutive mTORC1 activation, which we verified using immunoblots and in vivo lung sections staining. Early-onset inhibition of the mTORC1 pathway by in vivo rapamycin treatment (4–8-week-old mice) provided a mild protective effect by stabilizing the progression of the alveolar enlargement which correlates with the cytostatic effect of rapamycin observed in the clinical profile of LAM treatment[12].

Among the transcriptomic changes caused by the loss of *Tsc2* in lung mesenchyme, we found the upregulation of several

important molecular pathways including WNT, FGF, and TGFB (Fig. 4a) which could affect the formation of the observed phenotype. Moreover, we have found some of the key genes in these pathways to be regulated in an age- and sex-specific manner (Fig. 4b, c), suggesting the involvement of estrogen in their transcriptional regulation. In agreement with that, similar to that of observed in human LAM MACs, *Tsc2*<sup>KO</sup> MLFs had increased ERα-encoding *Esr1* gene expression (Fig. 4d). It has been previously shown that the ligand-independent trans-activation of the ERα specifically depends on S167 phosphorylation performed by

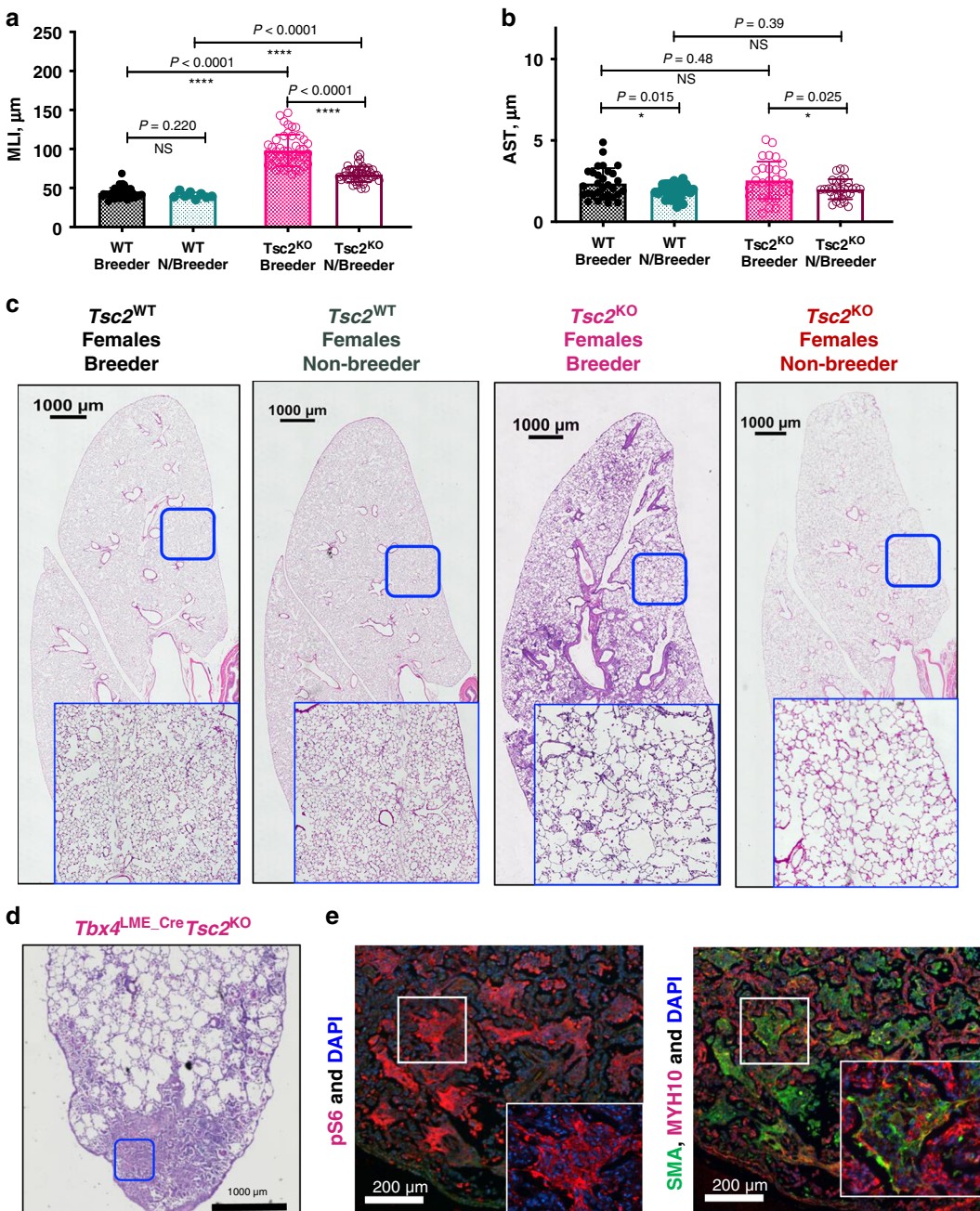

**Fig. 8 *Tsc2* KO in mouse lung mesenchyme induces LAM lung-like phenotype in female breeders. a** 54-week-old female breeders develop severe loss of alveolar structure with significantly higher MLIs than the non-breeder controls. $N = 4$ for each group. **b** Alveolar septal thickness is higher in breeders compared to non-breeders. Error bars for **a** and **b** represent mean values with SD. Statistical significance was determined using the Holm–Sidak method, with alpha = 0.05. Each row was analyzed individually, without assuming a consistent SD. Raw data for the graphs in panels **a** and **b** are supplied in the Source data file. **c** Representative images of the 54-week-old female lungs of breeders and non-breeders. $N = 4$ for each group. **d** Representative image of H&E stained sections of the paraffin-embedded lungs of the distal lung lesions developed in the 54-week-old breeder female mice. **e** Immunofluorescent staining of the distal lung lesions in 54-week-old female breeder mouse lungs shows positivity for pS6 (a marker of mTORC1 activation), as well as SMA and MYH10 (protein markers found in human lung LAM lesions).

S6K1[39], an enzyme that is hyperactive in TSC2-null cells, suggesting that it could lead to the synergistic upregulation of ERα transcriptional targets by simultaneous action of the estrogen and constitutively active mTORC1 pathway (Fig. 9). This could explain the increased expression of selected genes (e.g., *Wnt3a, Wnt5a, Fgf7*, etc.) specifically in female *Tsc2*KO MLFs and not female *Tsc2*WT MLFs.

Interestingly, our study revealed that *Tsc2* KO in MLF has affected global gene expression of multiple cell types including the

epithelium, by increasing the proliferative WNT response, and vascular endothelium by enhancing myogenic pathways (Supplementary Fig. 4c). We propose that such profound changes to the surrounding cell types could be induced by non-cell-autonomous stimulation originating from *Tsc2*KO MLFs. Using our model, we have shown the important role of WNT upregulation in *Tsc2*KO mouse lung mesenchyme and the interplay between WNT/β-catenin and mTORC1 signaling pathways in the age-dependent alveolar enlargement. It is known that select lung-

**Fig. 9 The scheme shows the cellular composition and structure of normal lung alveolus and the molecular changes occurring in *Tsc2*<sup>KO</sup> lung fibroblasts in female lungs.** Here, based on published data and the evidence presented in the manuscript, we illustrate the hypothetical mechanistic interplay of mTORC1, ERα, and WNT signaling pathways underlying the formation of the observed phenotype.

mesenchymal cell populations play a role in WNT/β-catenin signaling during lung regeneration[40,41]. Also, WNT stimulates translation and cell growth by activating the mTORC1 pathway by inhibiting GSK3-dependent phosphorylation of TSC2[42]. In our study, the deletion of TSC2 opened up a new opportunity to investigate the crosstalk between mTORC1 and WNT/β-catenin pathways. We found that the constitutive activation of WNT/β-catenin signaling was not sufficient to induce a LAM-like phenotype in mouse lung. However, the ablation of WNT/β-catenin signaling in the lung-mesenchymal cells combined with the TSC2-dependent mTORC1 activation prevented age-dependent alveolar enlargement in the lung parenchyma. With the addition of the ex vivo evidence presented in our alveolosphere assays, we speculate that the proteins in *Tsc2*<sup>KO</sup> MLF secretome may potentially impact AT2 cell fitness, specifically their role in the alveolar re-epithelization.

Collectively, based on the published data and our novel findings we created a hypothetical scheme of the intricate molecular interplay of the three pathways: mTORC1, ERα, and WNT in the development of LAM-like phenotype in our mouse model (Fig. 9). Due to the high degree of complexity of all three molecular pathways involved, a more detailed mechanism will have to be elucidated in future studies. Our mouse lung-mesenchymal *Tsc2*-null model will be useful in understanding the evolution of mTORC1-driven lung pathologies and will serve as a preclinical model for identifying and testing therapeutic targets, not only for LAM but also potentially for other lung diseases with activated mTORC1.

## Methods

**Immunohistochemistry**. Immunohistochemistry was performed following heat antigen retrieval methods and stained with the antibodies (the full list of antibodies can be found in the Supplementary Table 2). Human tissue samples from control and LAM lungs were obtained from human lung transplant donors signed informed consent, in accordance with the Declaration of Helsinki and approved by the National Disease Research Interchange (NDRI, Philadelphia, PA) and the PENN Lung Biology Institute Human Tissue Biobank in accordance with a protocol approved by the University of Pennsylvania Institutional Review Board.

**Human single-cell RNA-seq sample preparation**. Lung tissue samples were obtained from human lung transplant donors signed informed consent, in accordance with the Declaration of Helsinki and approved by the Institutional Review Board at the University of Pennsylvania and NDRI. Human LAM lung (ND_16635) and control lung (AGAH_121) samples from age-matched 70-year-old females were processed at the PENN Lung Biology Institute Human Tissue Biobank according to the protocol for human lung digest to gain single-suspension[32]. Briefly, visibly large blood vessels including arteries, arterioles, as well as bronchi and airways were dissected out to enrich the sample for parenchymal areas and exclude from the analysis of the abundant SMCs, lining the large blood vessels and airways. To decrease immune cell representation in total cell populations, immune cells were depleted according to the standard protocol of human CD45<sup>+</sup> cell depletion and then added back at 10% concentration. For single-cell RNA-seq analysis samples were prepared as follows: 18,000 of CD45<sup>neg</sup> cells combined with 2000 (10% add-back) of total CD45<sup>+</sup> cells. Samples were submitted

in a cell suspension of 20,000 cells for the cell single capture on 10X Genomics Chromium Chip.

**Single-cell RNA sequencing using In-Drop and the GemCode platform**. The cell suspension was loaded onto a GemCode instrument (10X Genomics)[43]. Briefly, single-cell barcoded droplets were produced using 10X Single Cell 3' v2 chemistry. Libraries generated were sequenced using the HiSeq Rapid SBS kit, and the resulting libraries were sequenced across the two lanes of an Illumina HiSeq2500 instrument in a High Output mode. Reads were aligned, and gene-level unique molecular identifier (UMI) counts were obtained using the Cell Ranger pipeline. Cell Ranger output results are presented in Supplementary Table 3.

All further analyses were performed using the Seurat v3.0 pipeline[44]. Briefly, the R package Seurat (version 3, http://satijalab.org/seurat/) was used to perform scRNA-seq analysis. Reads from the two sequencing repeats were pulled together, and cells were then filtered to have >500 detected genes and <5% of total UMIs mapping to the mitochondrial genome. The cell cycle phase score was calculated for each cell using the Seurat function CellCycleScoring. Data were scaled to remove unwanted variation from a number of genes, percent mitochondrial, cell cycle score reduced by regression. Principal component analysis (PCA) was used to create a reduced data set into a smaller number of components (eigengenes) while preserving the variation of the entire data set. The number of dimensions used in cluster analysis and dimension reduction procedures was determined using the JackStraw test. Dimension reduction was performed using the T-stochastic neighboring embedding method (t-SNE) or uniform manifold approximation protection (UMAP) method[45]. Seurat was used to create dimensionality reduction and Violin plots.

LAM lung samples represented a sporadic case of pulmonary LAM, and the depth of the scRNA-seq analysis did not allow us to detect any somatic mutations in the *TSC1*/*TSC2* locus of the analyzed lung cell subpopulations.

**Ligand-receptor analysis**. Ligand receptors gene pairs were obtained from the FANTOM5 project (http://fantom.gsc.riken.jp/5/suppl/Ramilowski_et_al_2015/). For a given cluster a ligand or receptor was considered expressed if 30% of cells had a UMI value of >0. A directed graph was constructed with the nodes as clusters and edges as a number of ligand receptors pairs between each cluster. R package igraph was used to make the network plots.

**Experimental animals**. All experiments involving animals conformed to the Guide for the Care and Use of Laboratory Animals published by the US National Institutes of Health (NIH Publication eighth edition, update 2011) and were approved by the Institutional Animal Care and Use Committee of the University of Pennsylvania. The studies were carried out in compliance with all ethical regulations. Mice were kept and observed by professional husbandry staff in the CRB vivarium. Rooms ranged from 68 to 78 F and 20–70% humidity. Lights are on 12 h cycle on 7a and off 7p year around.

*Tbx4*<sup>LME_Cre</sup> mice[26] were generously provided by Dr. Mark Krasnow, Stanford University. *Tsc2*<sup>loxP/loxP</sup> mice[46] were generously provided by Dr. Stephen Hammes at the University of Rochester. *Tsc2*<sup>loxP/loxP</sup> mice were crossed with *Tbx4*<sup>LME_Cre</sup> heterozygous mice to create *Tsc2*<sup>loxP/WT</sup>, *Tbx4*<sup>LME_Cre+/WT</sup> mice. The progeny was genotyped using primers P2F and P2R[46] for the *Tsc2* floxed transgene, and primers oIMR1084, oIMR1085, oIMR7338, and oIMR7339 designed by the Jackson Laboratory for generic *Cre* (see the Supplemental Table 2 for details). These heterozygous mice were then crossed to produce homozygous mice which would be the breeders for experimental mice. The mice referred to as *Tsc2*<sup>KO</sup> are *Tbx4*<sup>LME_Cre</sup> *Tsc2*<sup>loxP/loxP</sup>. The wild-type (WT) mice are *Tbx4*<sup>LME_Cre</sup> *Tsc2*<sup>WT/WT</sup>. Male and female adult (8- to 54-week-old) mice were used in the experiments. Morphological, morphometric, and immunohistochemistry analyses were performed on 8-, 12-, 20-, and 54-week-old mice. Pulmonary function tests were performed on 12- and 24-week-old animals.

*Tbx4*<sup>LME_Cre</sup>*Tsc2*<sup>KO</sup>*Ctnnb1*<sup>KO</sup> mice were generated by crossing *Tbx4*<sup>LME_Cre</sup>*Tsc2*<sup>loxP/loxP</sup> with *Ctnnb1*<sup>(exon2-6)loxP/loxP</sup> mice[31]. Morphometric

analysis was performed on 12-, 16-, and 20-week-old mice. $Tbx4^{LME\_Cre}Ctnnb1^{Ex3KO}$ mice were generated by crossing $Tbx4^{LME\_Cre}$ and $Ctnnb1^{(Ex3) \, loxP/loxP}$ mice[47]. Morphometric analysis for these mice was performed at 20 weeks of age.

**Preparation of mouse lungs for morphological, morphometric, and immuno-histochemistry analyses.** Mice were euthanized by a single dose of Euthanasia Solution (Pentobarbital based). The chest cavity was exposed, and the lungs cleared of blood by perfusion with cold PBS via the right ventricle. Lungs were inflated from control and experimental animals at a constant 25-cm $H_2O$ pressure, measured from the animal's chest[48]. Two-percent paraformaldehyde was used as the fixative. After inflation, the lungs were carefully dissected out and placed in a container full of 2% paraformaldehyde for fixation overnight at 4 °C. Lungs then went through a dehydration process at 4 °C with gentle rotation. The first step was 4 × 30-min PBS washes. Next, the lungs were placed in 30% ethanol for 2 h, then 50% ethanol for 2 more hours. The lungs were then left in 70% ethanol overnight. The next day the lungs were changed to 95% ethanol and left overnight. On the third day of lung dehydration, the lungs were placed in 100% ethanol and left overnight. The next morning lungs were placed in fresh 100% ethanol and stored at −20 °C for processing by the Histology Core at the University of Pennsylvania Cardiovascular Institute.

**High-throughput MLI and alveolar septal thickness (AST) analysis.** A software module has been developed with the Matlab 2018a environment (RRID: SCR_001622) using its image processing and statistical toolboxes to perform high-throughput mean length intercept (MLI) analysis of H&E stained sections of mice lung samples. Using this software the following steps had been performed: (1) reads were taken from multiple random sampled TIFF/JPEG images of H&E stained sections of PFA fixed mice lungs; (2) binarization was performed in an automated manner using global thresholding function that is based on Otsu's method; (3) the binarized images were processed using image operations such as erosion, dilation to digitally fill the capillary lumens in the alveolar walls; (4) images were inverted such that the alveolar spaces are black, and the walls are white; (5) the dimensions of the image were extracted and based on the vertical dimension, the equal spaced line grid consisting of 20 lines were created; (6) the number of times the lines intersect with the walls and calculate the length of the intercepts for each of lines based on dimensions of the image were counted; (7) then, a histogram of the intercepts were generated and computed the MLI, its associated standard deviation and dispersion index. The annotated codes for the high-throughput image analysis of mouse lung MLIs and ASTs can be found in the supplementary files and are available for public access in the GitHub repository: https://github.com/aravind245/biovision.

**In vivo rapamycin treatment.** Rapamycin treatment at 4 mg/kg started when $Tbx4^{Cre\_LME}Tsc2^{WT}$ and $Tbx4^{Cre\_LME}Tsc2^{KO}$ mice were 4 weeks old. Rapamycin was delivered via intraperitoneal injection (i.p.) three times a week (Monday, Wednesday, and Friday). The rapamycin was diluted to 1% DMSO in 99% PBS, so the vehicle group received injections with 1% DMSO in 99% PBS. Treatment started when mice were 4 weeks old and continued at the dosing mentioned above until the mice were 8 weeks old, at which point the mice were sacrificed and tissue collected for analysis.

**Mouse lung pulmonary function tests.** Invasive measurements of respiratory mechanics were performed under baseline conditions[49–51]. Briefly, mice were anesthetized with a solution of pentobarbital, tracheotomized, cannulated with a 20-gauge metal stub adapter (with a typical resistance of 0.39 cmH2O.s/mL) and connected to a small-animal, computer-controlled, piston ventilator (flexiVent; SCIREQ, Inc., Montreal, QC, Canada) for mechanical ventilation (150 breaths per min and a tidal volume of 10 mL/kg of body weight, a 2:3 inspiratory: expiratory ratio, and a positive end-expiratory pressure of 3 cmH2O.s/mL) and measurements. Following 2 deep lung inflations at 30 cmH2O to open-up closed lung spaces and standardize lung volume history, an automated series of measurement maneuvers assessing the mechanical properties of the subject's respiratory system was repeated three times for each mouse. The parameters obtained from this sequence, integrated by default in the operating software (flexiWare v.7.6.4), were averaged for each subject and group.

**Mouse lung lineage separation.** Cell lineage separation from mouse lung was generated as follows. First, the lung was removed and minced with a razor blade. Minced lung was placed in a digestion solution containing 480 U/ml Collagenase Type I (Life Technologies) 50 U/ml Dispase (Collaborative Biosciences) and 0.33 U/ml DNase (Roche), this was allowed to incubate in a 37 °C water bath with frequent agitation for 45 min. The cell solution was filtered through 100 and 40 mm cell strainer (BD Falcon). ACK lysis was used to remove blood cells. Cell pellets were resuspended in FACS buffer, containing sterile PBS with 1% FBS and 1 mM EDTA. Antibodies used for magnetic bead sort were biotinylated anti-CD45, EpCAM, CD31, PDPN, and PDGFRa (BioLegend, San Diego, CA). Full information about the antibodies is presented in Supplementary Table 2.

**Mouse cell population bulk RNA-seq (pop-Seq).** Library prep was conducted using Illumina truSeq stranded mRNA kit and Clontech SMARTer RNA-seq amplification kit. Fastq files were assessed for quality control using the FastQC program. Fastq files were aligned against the mouse reference genome (mm9) using the STAR aligner[52]. Duplicate reads were flagged using the MarkDuplicates program from Picard tools. Per gene read counts for Ensembl (v67) gene annotations were computed using the R package with duplicate reads removed. Gene counts represented as counts per million (CPM) were first nominalized using the TMM method in the edgeR R package and genes with 25% of samples with a CPM < 1 were removed and deemed low expressed. The data were transformed using the VOOM function from the limma R package[53]. Differential gene expression was performed using a linear model with the limma package. Given the small sample size of the experiment, we employed the empirical Bayes procedure as implemented in limma to adjust the linear fit and calculate P values. P values were adjusted for multiple comparisons using the Benjamini–Hochberg procedure. Heatmaps and PCA plots were generated in R.

**Western blot analysis.** Cell monolayers were washed once with DPBS, then lysed on ice for 15 min in 100–200 µl RIPA cell lysis buffer (Sigma) supplemented with protease and phosphatase inhibitors (Roche). Protein was determined by the BCA (Thermo Scientific) assay and equal amounts of proteins resolved by SDS PAGE on 3–8% Tris-Acetate (large proteins), 4–12% Bis-Tris or 10–20% Tris-Glycine Novex gels (small proteins) (Thermo Fisher), transferred to nitrocellulose via iBLOT, blocked with TBS LiCor Blocker and incubated overnight at 4 °C with the primary antibody diluted appropriately in Blocker: 0.1% Tween 20. Blots were washed with TBS: 0.1% Tween 20 then incubated for 60 min RT with secondary antibody LiCor 800W-labeled diluted 1:15,000 in Blocker: 0.2% Tween 20. Blots were rewashed, dried and fluorescent image acquisition and band intensity quantification performed using an Odyssey IR imaging system (LiCor Biosciences Lincoln, NE). Antibodies used for immunoblots were b-Actin, pS6 (S235/236), TSC2, mTOR, pmTOR (ser2448), S6, 4EBP1, p4E-BP1 (Thr37/46), 4EBP1, pMNK (Thr197/202), MNK, pAkt (Ser473), Akt, and β-Actin were all from Cell Signaling. Antibody to peIF4E (Ser209) (76256) was from Abcam. All primary antibodies were at 1:1000 dilution except β-Actin which was 1:15,000 dilution. The full list of antibodies can be found in Supplementary Table 4.

**Alveolar organoid assay.** Alveolar organoid assays were performed as follows[54]. To obtain SPC-GFP primary cells, $Sftpc$GFP mice were euthanized, and single-cell suspensions were made from the lung tissue as described above. Following red blood cell lysis with ACK buffer the cell suspensions were filtered through 40 µm cell strainers (BD Falcon), counted and resuspended in FACS buffer for the FACS sort of GFP+ cells. Sorted cells were collected in DMEM + 10%FBS, after which cells were centrifuged and counted by trypan exclusion. Mesenchymal support cells were isolated from $Tbx4^{Cre}Tsc2^{WT}$ or $Tbx4^{Cre}Tsc2^{KO}$ mouse lungs according to the sorting procedure depicted in Supplementary Fig. 4a. Approximately $5 \times 10^3$ SPC cells were mixed with $5 \times 10^4$ mesenchymal cells in 50% Matrigel (growth factor reduced, phenol-red free) (Corning) in the small airway growth media (SABM, Lonza) with the following additives: 1x insulin/transferrin, 0.1 mg/ml Cholera Toxin (Sigma), 25 ng/ml EGF (Peprotech), 30 mg/ml bovine pituitary extract (Lonza), 0.01 mM Retinoic acid (Sigma & Lonza), and 5% FBS (Denville). Rock inhibitor, Y27632 (Sigma) was included in the media for the first 2 days and fresh media was added every 2 days. Ligand treatments of organoids were performed using the following reagents at the indicated concentrations, FGF7 25 ng/ml (R&D Systems), E2 (Sigma), rapamycin (Sigma). Ligands were added at the time of the first media change (after removal of Rock inhibitor), new ligands were added upon each media change or every other day. Organoids were cultured for 21 days, then imaged and harvested.

**qPCR analysis of gene expression.** Total RNA was isolated from cells using the RNeasy Mini kit (Qiagen, Cat #74174). RNA samples were converted to cDNA using the SuperScript™ IV First-strand synthesis system (Invitrogen, cat#18091050). Quantitative PCR was performed in triplicates using StepOne Real-time PCR System (Applied Biosystems), SYBR-green Fast master mix (Applied Biosystems, #4385612), and the primers specific to the gene of interest (Full list of primers can be found in the Supplementary Table 4). Differential expression was calculated as a fold-increase using ΔΔCt method with normalization to beta-actin.

**Statistical analysis.** Statistical analysis was performed in GraphPad Prism software. For PFT test analysis two-tailed parametric T-test was used for the comparison between two experimental groups, and a one-way ANOVA was used for multiple comparisons. Data were considered significant when $p < 0.05$. Lung function results were analyzed using a two-way ANOVA with Sidak's multiple comparison test. The results are represented as mean $+/−$ standard deviation. In all other grouped comparisons multiple T-tests were used. Statistical significance was determined using the Holm–Sidak method, with alpha = 0.05. Each row was analyzed individually, without assuming a consistent SD.

**Reporting summary**. Further information on research design is available in the Nature Research Reporting Summary linked to this article.

## Data availability

Human scRNA-seq and bulk mouse lung cell population RNA-seq data that support the findings of this study have been deposited to the public functional genomics data repository Gene expression Omnibus (GEO). Datasets have a "super-series" accession with the accession code GSE139819. The Source data file is supplemented with the manuscript. Any remaining data supporting the results of the study will be made available from the corresponding author upon reasonable request. Source data are provided with this paper.

## Code availability

The annotated codes for the high-throughput image analysis of mouse lung MLIs and ASTs are available for public access in the GitHub repository: [https://github.com/aravind245/biovision]. The rest of the source data that support the findings of this study are available from the corresponding author upon reasonable request.

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

## Acknowledgements

We are very grateful to LAM patients for their donation of tissues and world-wide LAM Foundations for their sense of urgency and advocacy for the development of new therapies for LAM. We thank the NDRI for providing LAM tissue samples used in immunohistochemical analysis. We thank Dr. Steve Albelda, Dr. Jarod Zepp, and Dr. Slaven Crnkovic from Ludwig Boltzmann Institute for Lung Vascular Research, Graz, Austria for helpful discussions and reading of the manuscript. Dr. Jarod Zepp generously provided an original drawing used in Fig. 9. Krymskaya Lab is funded by the National Institutes of Health (HL151467, HL13126, HL141462, HL114085, HL110551, and UO1HL131022). The Morrisey Lab is funded by the National Institutes of Health (HL087825, HL134745, HL132999, and HL132349) and the Longfonds BREATH Lung Regeneration Consortium. M.C.B., S.M.L., and J.K. Fellowships are funded by NIH/NHLBI T32 HL007586.

## Author contributions

K.O. designed and performed the experiments, mouse population RNA-seq, downstream bioinformatic analysis of scRNA-seq and mouse RNA-seq, qPCR validation, morphometric and quantitative immunohistochemical analysis, analyzed and interpreted the data, and wrote the manuscript. M.C.B. developed the protocol for human scRNA-seq lung sample preparation, performed scRNA-seq preparation, and participated in scRNA-seq data analysis and interpretation. R.R. performed breeding and genotyping of experimental animals, animal treatments, lung isolation, tissue preparation for immunohistochemistry, morphometry, and pulmonary function tests. A.S. designed and developed a software module used for morphometric analysis of mouse lung. S.M.L. performed immunohistochemical analysis. A.R.M. performed immunohistochemical staining and analysis of human and mouse lung tissue and prepared IHC figures. A.G. participated in the morphometric analysis. J.F.E. performed western blot analysis, participated in data analyses, writing the manuscript. M.K. performed pulmonary function tests. K.O., A.R., J.K., M.F.B., J.E., and V.P.K. analyzed and interpreted pulmonary function data. E.A.V. and K.O. performed MLI measurements. A.S., D.B.F., and K.O. performed high-throughput MLI analysis. S.L. performed initial organoid assay. K.O. performed organoid assays, organoid section staining analysis, colony-forming efficiency. J.C. isolated single cells from the human lungs. A.B and M.M. performed bioinformatics analysis. E.C. performed lung resections and provided patient lung samples. E.E.M. led the scRNA-seq analysis, interpreted the data, provided input for conceptualizing, and writing the manuscript. V.P.K. designed the study, interpreted the data, wrote the manuscript with input from co-authors, and directed the project.

## Competing interests

The authors declare no competing interests except for A.R. who is employed by SCIREQ Inc., a commercial entity involved in a subject area related to the content of this article. SCIREQ is an emka TECHNOLOGIES company.
