## [Peer Review File · Nature Communications]

Reviewers' comments:

Reviewer #1 (Remarks to the Author):

This paper seeks to throw light on the etiology of the abnormal lung phenotype in patients with lymphangioleiomyomatosis (LAM), a rare progressive disease in women involving overgrowth of smooth muscle-like cells and abnormal epithelial cells in a cystic lung. In some case LAM arises in patients homozygous for null mutations in genes of the tuberous sclerosis complex (TSC1 and TSC2, associated with overactivity of mTORC1. There are two main parts to the analysis: single cell RNAseq data on ~7-10K cells from each of the lungs of 2 normal female donors and 2 female LAM patients and analysis of an adult mouse model in which Tsc2 is conditionally inactivated in Tbx4+ mesenchymal cells of the lung. The authors conclude that the LAM phenotype is due, at least in part, to defects in the normal reciprocal trophic interaction between alveolar type 2 stem cells and a specific subpopulation of mesenchymal cells present in their niche (cells they designate as mesenchymal alveolar niche cells or MANCs) which results in the accumulation of different populations of abnormal cells (LAM cells), that they argue are derived from MANCs, and either AT2 cells or some cell type intermediate between AT2 and AT1, The questions under investigation are clinically important and the resources are appropriate. However, the interpretation of some of the patient and experimental data is very unconvincing and the overall take home message is confusing and too preliminary to merit publication at this time.

Specific comments

Interpretation of the single cell RNAseq data and the origin of abnormal mesenchymal and epithelial LAM cells

1. What was the relevant genotype of the 2 female LAM patients i.e. did they have mutations in TSC1/2?? Were the lungs mosaic for homozygous mutations? Is it possible that only a proportion of the cells isolated for single cell analysis were homozygous? If so, this needs to be discussed upfront.
2. The authors focus on a population of mesenchymal cells in the human lung that they term "MANCs" (mesenchymal alveolar niche cells). However, this is not a universally accepted designation, but one proposed by the Morrissey lab for a subpopulation of mesenchymal cells in the mouse lung based on a combination of gene expression data, spatial location near AT2 cells, and function in an organoid assay (Zepp et al 2017 i.e. they are not just defined by the expression of a few genes, as claimed on page 6 line 77/78). For the current paper to have broad impact the authors need to show clearly: (a) How their mouse MANCs relate to alveolar mesenchymal cell populations in the vicinity of AT2 cells reported by others; (b) The spatial location of their human "MANCs" in the intact lung and (c) How their human "MANCs" relate to populations of mesenchymal cells in the human lung reported by others from single cell RNA seq studies (e.g. Reyfman et al AJRCCM 2019

“Single-cell transcriptomic analysis of human lung provides insights into the pathobiology of pulmonary fibrosis”; Travaglini et al 2019 bioRxiv <http://dx.doi.org/10.1101/742320>. “A molecular cell atlas of the human lung from single cell RNA sequencing”. In other words, is their human cell population similar to “alveolar fibroblasts” or to “lipofibroblasts” of Travaglini et al. If neither, then readers need to have a clear and upfront discussion of the issues.

3. Fig 1b and c are very hard to interpret. Where are SMCs in the normal lung? Can the authors be sure that the AT2/AT1 cell population is not doublets? What is the evidence that they represent a “transitional” cell population? Are they cells present in mouse lung undergoing repair after injury or compensatory regrowth after partial pneumonectomy, for example?

4. Where are lymphatic endothelial cells and how are “LAM” cells related to lymphatic endothelial cells?

5. In Fig 1e There seems to be no arrow for crosstalk between MANCs and AT2 cells in the control lung yet interaction between these populations is surely one of the defining characteristics of MANCs in Zepp et al?

6. On page 5, lines 70/71 the authors bluntly state that “AT2 cells serve as bipotent progenitors”. This claim is not supported by the experimental data in the references given. The following sentence beginning “It is interesting to note....” is not supported by the reference given (18), which is about extrapulmonary LAM.

Alveolosphere assay for evidence of aberrant epithelial-mesenchymal interactions in the LAM lung

1. The alveolosphere assay used here is not well documented. It appears that total “mouse lung fibroblasts” (MLF) are used and not a subpopulation of mesenchymal cells as described in Zepp et al. In the earlier population the authors claimed that “MANCs” were the most efficient at supporting the organoid assay so why was this population not used for these assays?

2. The organoids are very small and do not contain robust populations of AT1 cells. We are not told how long they are cultured before harvest and analysis

3. Why does the number of colonies increase over time and not plateau in Fig 3e? Is colony number a good measure of organoid “growth speed” (rather than colony diameter or circumference, for example?) In Fig 3g ,when were the colonies counted?

4. Are the differences shown in Fig 3f graph statistically significant? In Fig 3h there are no error bars. Only 2 biological replicates were done and again there is no indication of significance.

Reviewer #2 (Remarks to the Author):

In this manuscript by Obratsova K et al., authors identified and characterized lymphangioliomyomatosis (LAM) cells by single cell RNA-seq (scRNA-seq) of patient and age/sex-matched normal lung tissues. Signature of LAM cells were also validated by biochemical and histochemical analysis on mouse model of mesenchymal-restricted deletion of Tsc2 in lung. Authors concluded that LAM cells originated/differentiated from mesenchymal alveolar niche cell (MANC) might form new cellular interaction between LAM and epithelial cells for the development of LAM-like pathologies. This is an exciting piece of work yet I have several comments and suggestions about the interpretation of the results that may help improve the study. I detail them below:

1. Authors identified LAM cells from parallel comparison of UMAP analysis on normal and patient scRNA-seq data. It might be originated from batch effect of two experiments from two different individuals. I ask authors to show cell type cluster data with aggregated samples from patient and normal lung together.
2. Is there any possibility of cellular interaction between LAM or MANC with immune cells such as alveolar macrophages for the pathogenesis of LAM.
3. Is there any difference in cell type proportion? For example, immune cell cluster of normal lung in Fig 1a looks bigger than that of patient lung.
4. What is the immune cell cluster in Fig 1a? Authors used CD45-depleted cells in single cell profiling, but still it is not clear.
5. Authors used large cell numbers for single cell analysis, but sequencing read number for single cell is relatively low. Is there any reason to use low sequencing depth?
6. In Fig 1e and 1f, authors described cellular interactions in LAM and normal lung. However, it was based on co-expression of receptor and ligand in LAM lung, so it requires to be investigated meticulously with validated methods such as CellPhoneDB.
7. What is the role of TSC gene deletion in LAM cells? Please show the changes in mTORC signaling and status of TSC1/TSC2 in single cell of LAM patient.
8. In addition, LAM-specific marker can be validated in patient tissue. Visualization of LAM cells and increased population of LAM/MANC cells in patient tissue might provide more convincing evidence.

Reviewer #3 (Remarks to the Author):

In this manuscript, Obratsova and colleagues highlight the importance of TSC2 deletion in lung mesenchymal cells. This is a thorough paper, with a logical progression from transcriptome analysis to in vivo experiments driven by the interpretation of the RNA sequencing results, which was well thought out. This work expands our understanding of the mesenchymal origin of LAM, demonstrating the mesenchymal and transitional alveolar epithelial states of LAM. It shows that mTORC1 activation via loss of TSC2 in lung mesenchymal cells drives “LAM phenotype” and has elegantly developed a mouse model of LAM. The major weakness of the work, however, is that while a novel mouse model of LAM that sheds light on origin of LAM was generated, clear mechanisms were not identified.

Major comments:

1. Rapamycin had a partial inhibitory effect in the 3D organoid from TSC-KO mesenchymal cells (Figures 3g and h); and rapamycin withdrawal led to the increase in size of the 3D organoid colonies. It would be interesting to see if pS6, SMA, MYH10 also decreased with rapamycin and increased with rapamycin withdrawal. This will help demonstrate that mTOR activation is the key pathway in this system.

2. In this paper, the authors stated that LAM phenotype with TSC2 KO in lung mesenchymal cells, is correlated with upregulation of WNT signaling pathway, (WNT ligands, Figure 4c).

In order to show this as a mechanism, it would be beneficial to see if increased expression of WNT ligands in normal lung mesenchymal cells would develop into LAM phenotype. Also, if rapamycin inhibits the up-regulation of WNT ligands.

3. The paper interestingly shows that mTOR activation due to TSC2 loss activates ER and its target genes (extended Figure 7A-C), which is a very novel finding. They showed that gender-specific and age specific LAM phenotype with decline in LAM-lung function more in females, more so in breeders. They also showed that mTOR activation due to TSC2 loss increases WNT ligands in lung mesenchyme that activates WNT signaling pathway. However, there is a lack of mechanistic connection between mTOR, WNT signaling and ER pathways with the LAM phenotype.

4) Extended Fig 2B: Since AT2 signature genes are expressed in the AT1 gene clusters and vice versa in the control lungs, the category saying AT1/AT2 needs a little more definition, like how many fold increase in both AT1 and AT2 genes. (Same for Fig 1 d)

5) Figure 2d: This figure shows that WNT signaling is increased by TSC2 KD in lung mesenchymal cells, shown in the next figure by increase of WNT ligands.

Further experiments showing the importance of WNT signaling pathways as an important pathway in LAM would be helpful. For example, experiments showing increased expression of WNT ligands in normal lung mesenchyme lead to LAM phenotype. Or, experiments showing that inhibition of WNT signaling pathways reverses the LAM phenotype.

Minor comments:

1) Fig. 1: They don't say how many subjects were part of this?

2) Fig. 2a: Tsc2 expression is not quite as gone in the KO compared to whole lungs?

3) Figure 3A: the genes described in the text (lines 167-168) refer to Fig 3A, which do not show the genes Wnt3a, Wnt4, Wnt5a, and Wnt10b. although qPCR results are shown in extended Fig 7)

4) Figure 3b: gene Fzd2 is not shown in the array although mentioned in the text (line184)

5) Fig. 3d, g, h, j: how many mice/experiments used/performed?

6) Extended Fig. 1: please spell out MYH-10 (myosin heavy chain-10) in the legend.

7) Extended Fig 5A: What does B, V and B stand for? Its not written in the figure legend

8) Extended Fig. 6B: number of mice used within each group not stated.

9) Extended Fig. 7A: What is the p-value comparison between the male and female knockouts in all of these groups as related to upregulation in ERalpha regulation to underscore sex as a biological variable?

10) The overall organization of the manuscript into many extended figures that are referred to in the text out of order is very confusing. The authors may consider re-organizing the presentation of their data in a progressing order.

11) The Discussion is very descriptive, many of the details mentioned belong in Introduction. It would have been better if the authors placed their findings in the context of the disease, focused on interpretation or the results, and speculated on the underlying mechanisms behind their observations. Specifically, would be interesting to discuss future direction for this TSC2 model, as well as what new therapies are a possibility based on the findings.

RESPONSE TO REVIEWER #1

We greatly appreciate the critical review by Reviewer #1 and encouraged by his/her opinion that “the questions under investigation are clinically important.” As recommended by the Reviewer’s valuable criticism, we performed new experiments and included new data in the revised manuscript. We also extensively revised the manuscript and explained the data based on insightful comments of the Reviewer to improve the data and to have “a broad impact of the manuscript.” Point-by-point responses are presented below.

Specific comments

Interpretation of the single-cell RNAseq data and the origin of abnormal mesenchymal and epithelial LAM cells

“1. What was the relevant genotype of the 2 female LAM patients i.e. did they have mutations in TSC1/2?? Were the lungs mosaic for homozygous mutations? Is it possible that only a proportion of the cells isolated for single-cell analysis were homozygous? If so, this needs to be discussed upfront.”

RESPONSE: The patient’s clinical history for the LAM lung samples used in our study, specifically the lack of family history of LAM or TS and records of the genotyping data for *TSC1/2* mutations, indicates a sporadic LAM. In sporadic pulmonary LAM, exonic mutations of *TSC1/TSC2* genes are not commonly seen¹, or the mutations could be mosaic and not easily detectable². Following the comparative analysis of the cell types in the LAM lung and control human lung, we have identified a unique cell subtype in the LAM lung which did not have an equivalent in the control healthy lung which we named “LAM lung cells”. We are aware that the term “LAM cells” has been historically used to define TSC2-negative cells. However, we were unable to look for point mutations or to show the loss of *TSC1* or *TSC2* gene expression in the newly identified LAM lung cell subset, nor the other cell types in the LAM lung due to insufficient depth of the scRNA-seq method. Subsequently, we refrained from calling this cell subtype LAM cells, but instead, we called it “LAM lung cells” to emphasize its unique origin. We agree with the reviewer that it is important to provide clarification about the status of *TSC1/2* mutations. We have included substantial clarification of this matter in the discussion of the revised manuscript.

“2. The authors focus on a population of mesenchymal cells in the human lung that they term “MANCs” (mesenchymal alveolar niche cells). However, this is not a universally accepted designation, but one proposed by the Morrisey lab for a subpopulation of mesenchymal cells in the mouse lung based on a combination of gene expression data, spatial location near AT2 cells, and function in an organoid assay (Zepp et al 2017 i.e. they are not just defined by the expression of a few genes, as claimed on page 6 line 77/78).”

RESPONSE: We agree with the Reviewer’s opinion and addressed this criticism by removing this abbreviation and revised the text and figures accordingly throughout the manuscript.

“For the current paper to have a broad impact the authors need to show clearly: (a) How their mouse MANCs relate to alveolar mesenchymal cell populations in the vicinity of AT2 cells reported

by others;” (b) The spatial location of their human “MANCs” in the intact lung and (c) How their human “MANCs” relate to populations of mesenchymal cells in the human lung reported by others from single-cell RNA seq studies (e.g. Reyfman et al AJRCCM 2019 “Single-cell transcriptomic analysis of human lung provides insights into the pathobiology of pulmonary fibrosis”; Travaglini et al 2019 bioRxiv, <http://dx.doi.org/10.1101/742320>. “A molecular cell atlas of the human lung from single-cell RNA sequencing”. In other words, is their human cell population similar to “alveolar fibroblasts” or to “lipofibroblasts” of Travaglini et al. If neither, then readers need to have a clear and upfront discussion of the issues.”

RESPONSE: We greatly appreciate the Reviewer’s valuable suggestion to improve our manuscript to have a broad impact by defining the way our mesenchymal cells relate to the mesenchymal cell populations discovered by Travaglini et al.³ and by Reyfman et al.⁴. As suggested by the Reviewer, we compared differential gene expression patterns of the mesenchymal subtypes detected in our study to the human lung stromal cell types described in Travaglini et al.³, as well as to the fibroblasts from IPF lungs described in Reyfman et al.⁴. Our population of lung fibroblasts (previously termed MANCS) is most similar to the “alveolar fibroblasts” and “adventitial fibroblasts” described in Travaglini et al. based on the similarity of the gene marker expression, specifically *SPINT2*, *FGFR4*, *ITGA8*, *SFRP2*, and *SERPINF1*. We revised the text of the results and the discussion sections accordingly and re-named this population as Mesenchymal Alveolar Cells (MACs). The sub-population previously named “Fibrotic MANCs” in a similar comparison appeared to be comparable to the fibroblasts identified in the IPF lungs in Reyfman et al.⁴, based on the expression of *COL3A1*, *COL1A1*, and *ACTA2* genes. We re-named this subpopulation as fibrotically-activated MACs, since they retained the core gene marker expression of original MACs, while additionally expressing fibrosis-specific genes.

“3. Fig 1b and c are very hard to interpret. Where are SMCs in the normal lung?”

RESPONSE: To address the Reviewer’s comment and to answer the Reviewer’s question on where the SMCs cluster in the normal lung is, we provide clarification on the special way we have prepared our lung samples for scRNA-seq analysis. Prior to the single-cell analysis, we micro-dissected the major airways and vessels in the control and LAM lungs. We chose this approach (1) based on the existing knowledge of the abnormal growth of the immature-looking smooth muscle-like LAM cells within the lung parenchyma of the LAM lung, and (2) to focus our cell type analysis on the parenchymal lung areas while excluding from our comparison all other rather abundant vascular and airway SMCs.

As shown in **Extended Data Fig. 3**, comparative analysis of the mesenchymal cell subtypes of the lung parenchyma identified a sub-cluster of SMCs only in the LAM, but not in control lung sample. This cluster is shown as a differentiated SMCs (dSMCs) cluster in **Figs. 1c** and **1d** and is defined by the enhanced expression of differentiated SMC gene markers, including *MYLK*, *MYL9*, *DES*, but lacking *PDGFRβ* expression, a marker of alveolar myofibroblasts (AMFs). We included this explanation in the revised text of the Results and Methods sections.

“Can the authors be sure that the AT2/AT1 cell population is not doublets? What is the evidence that they represent a “transitional” cell population? Are they cells present in mouse lung undergoing repair after injury or compensatory regrowth after partial pneumonectomy, for example?”

RESPONSE: In response to the Reviewer’s comment we performed additional analysis of scRNA-seq cell clusters using the program DoubletFinder⁵ which classifies each cell as a singlet or a doublet. Based on this analysis cluster 16 (AT2/AT1) in the LAM lung samples returned a relatively low doublet score – 21%. However, to prove the existence of the transitional AT2/AT1 cell type *in vivo*, we performed double immunofluorescent staining of lung sections from multiple control and LAM lungs using cell-type-specific antibodies – HTII-280 for AT2 cells and HTI-56 to detect AT1 cells⁶. Quantitative analysis of the series of confocal images identified approximately 3.5% of total alveolar epithelial cells (AECs) in control lung to be double-positive for AT2 and AT1 markers. In contrast, LAM lungs showed on average 17.5% of AECs to be AT1/AT2 transitional cells. These new data are displayed in the revised **Figure 1, panels e and f**, and described in detail in the revised text of the manuscript.

“4. Where are lymphatic endothelial cells and how are “LAM” cells related to lymphatic endothelial cells?”

RESPONSE: We have identified lymphatic endothelial cell (LEC) clusters in both control and LAM lung samples based on the high differential expression of *LYVE1*, *PROX1*, *PDPN*, and *CCL21* genes. These data, as well as the detailed description of all single-cell clusters, were not included in the manuscript to keep the focus of the study on the differences in cellular profiles between the healthy control and LAM lungs, and to specifically highlight mesenchymal-epithelial interactions in the alveolus. However, we understand and appreciate the Reviewer’s insightful comment regarding the relatedness of the LAM lung cell cluster to LECs. To answer this question, we provide additional data below: the figure illustrates side by side comparison of the gene expression patterns between MACs, LECs, and LAM lung cells. The heatmap is based on the top 100 differentially expressed genes in the LAM lung cell cluster (the full heatmap including a comparison to all cell clusters in LAM lung is shown in **Extended Data Fig. 4A**). As evident from this comparison, the LAM lung cell cluster exhibits transcriptional pattern which is the most similar to MACs. However, there’s some relation of the LAM lung cell cluster to LECs as well, with comparable differential expression of the select genes, namely *BCHE*, *MEDAG*, *CD200*, *PROCR*, *CDC80*, *MAF*.

“5. In Fig 1e There seems to be no arrow for crosstalk between MANCs and AT2 cells in the control lung yet interaction between these populations is surely one of the defining characteristics of MANCs in Zepp et al?”

RESPONSE: We appreciate the Reviewer’s comment. Indeed, Zepp et al. established the interaction between MANCs and AT2 cells in the mouse lung as one of the defining characteristics of MANCs, and it is a valid point to look for such interaction in the crosstalk diagram depicted for our control human lung samples. As mentioned above, we will refrain from using the term MANCs regarding the mesenchymal alveolar cells in this work, since we are lacking spatial and functional evidence to make such a comparison. We did get a number of interactions predicted for MACs → AT2 axis both in the control and LAM lungs based on FantomDB ligand-receptor analysis. However, in the control lung, the number of interactions on this axis is substantially lower and got filtered out in the process of creating the diagrams shown in **Fig. 1e (revised**

Fig.1g). These diagrams were meant to show the contrast in the cellular crosstalk patterns; therefore, they depict only the most robust interactions in each sample.

“6. On page 5, lines 70/71 the authors bluntly state that “AT2 cells serve as bipotent progenitors”. This claim is not supported by the experimental data in the references given. The following sentence beginning “It is interesting to note....” is not supported by the reference given (18), which is about extrapulmonary LAM.”

RESPONSE: Thank you for the comment. We revised the text on page 5 and included new references by Nabhan, Science, 2018⁷, and Zacharias et al., Nature, 2018⁸. We removed the sentence containing reference 18 in the original manuscript.

“Alveolosphere assay for evidence of aberrant epithelial-mesenchymal interactions in the LAM lung

- 1. The alveolosphere assay used here is not well documented. It appears that total “mouse lung fibroblasts” (MLF) are used and not a subpopulation of mesenchymal cells as described in Zepp et al. In the earlier population the authors claimed that “MANCs” were the most efficient at supporting the organoid assay so why was this population not used for these assays?”*

RESPONSE: Indeed, as noted correctly by the Reviewer, we used total mouse lung fibroblasts and not MANCs. The scope of this study was to determine whether *Tsc2* loss in *Tbx4*⁺ mesenchymal lung progenitors was sufficient to produce sex- and age-specific LAM phenotype. We also aimed to characterize the effect of *Tsc2*^{KO} in lung mesenchyme on the fitness of the surrounding cell types, including AEC cells, the ratio of which was shifted in *Tbx4*^{LME-Cre}*Tsc2*^{KO} lungs. To answer those questions, we had to use the total population of *Tsc2*^{KO} mesenchymal cells since they originate from *Tbx4* lineage which marks multiple sub-populations of the lung mesenchyme. Our data demonstrate that conditional inactivation of *Tsc2* in *Tbx4*⁺ mesenchymal cells of the lung induces the LAM phenotype, at least in part, due to the defects in normal reciprocal trophic interaction of *Tsc2*^{KO} cells with AT2 cells. In our future studies, we plan to investigate which specific subtype of *Tsc2*-null *Tbx4*⁺-derived cells affects the LAM phenotype development the most and is the most efficient at stimulating organoid growth. To satisfy the Reviewer's comment, we provided more details about the alveolosphere assays used in our study and clarified which cells were used as a support for AT2 cell growth in the revised text of the manuscript.

“2. The organoids are very small and do not contain robust populations of AT1 cells. We are not told how long they are cultured before harvest and analysis”

RESPONSE: Thank you for this question. We described in the Methods section that the organoids were cultured for 21 days before the harvest and analysis. To address this comment by the Reviewer, we revised the legend to **Fig. 3**. In response to the Reviewer's comment about the size of the organoids, we compared our data to the ones published in Zepp et al.⁹ where AT2 organoids were grown on 5 various mesenchymal lineages. Our assay showed a size range of

~50uM, which was comparable to that of described for the unsorted mesenchymal support cells (labeled “others”) in Zepp et al.⁹ Regarding the populations of AT1 cells, we have to respectfully disagree with the Reviewer. As shown in **Fig. 3f**, the total population of *Tsc2*^{KO} mouse lung fibroblasts (MLFs) significantly increased the AT1 cell populations within analyzed individual organoid colonies.

“3. Why does the number of colonies increase over time and not plateau in Fig 3e? Is colony number a good measure of organoid “growth speed” (rather than colony diameter or circumference, for example?) In Fig 3g, when were the colonies counted?”

RESPONSE: We appreciate the Reviewer’s question. Colony number or commonly used term “colony-forming efficiency” (CFE) reflects the ability (or potential) of the single AT2 cells to form colonies in a 3D Matrigel environment when co-cultured with lung mesenchymal cells. In our alveolosphere assay, we aimed to compare CFEs of the AT2 organoids supported by mesenchymal cells that originated from *Tbx4*^{LME-Cre}*Tsc2*^{KO} and *Tbx4*^{LME-Cre}*Tsc2*^{WT} mouse lungs as a function of time, i.e. speed of colony formation. Given that each cell type (AT2 and lung mesenchymal cells) is plated in the 3D matrix under equal cell density and maintained under the same experimental conditions, this approach is the most basic and reliable measurable. The time points were taken until day 21, a time-point selected according to a well-established alveolosphere assay protocol by B. Hogan’s group¹⁰. Due to the goal of our study to compare the effects of *Tsc2*^{KO} and *Tsc2*^{WT} MLFs on the AT2 cell growth and trans-differentiation *ex vivo*, we considered the colony size (circumference or diameter) to be a measure of a given time point, because of its dependence on the external parameters such as the density of the matrix and mesenchymal support cells. These parameters may change unevenly as a function of time (for example, *Tsc2*^{KO} MLF may not grow with the same speed as WT MLF and will densify the collagen matrix with a different efficiency). Therefore, we decided to take colony size measurement at a terminal time point (21 days) to highlight the conditional differences, as depicted in **Figure 3h** (former **Figures 3g and 3h**). We included these details in the text and figure legend of the revised manuscript.

“4. Are the differences shown in Fig 3f graph statistically significant? In Fig 3h there are no error bars. Only 2 biological replicates were done and again there is no indication of significance.”

RESPONSE: In response to the Reviewer’s comment, we revised **Figs. 3f and 3h**: **Fig. 3f** now includes the P-values of the pairwise comparisons, **Fig. 3f** is now a part of **Fig. 3h**, since it illustrates the data from the same assay (with the two graphs showing the number and size of the organoids depending on treatment and on the type of mesenchymal support). As the Reviewer rightfully mentioned, the experiment in **Fig. 3h** was performed in 2 biological replicates (meaning that the support MLFs originated from 2 different WT and 2 different *Tbx4*^{LME-Cre}*Tsc2*^{KO} mouse lungs). However, it was repeated in 2 technical replicates for each biological replicate and each treatment condition. The error bars and the statistics were calculated for a combined total of 4 repeated measurements for each condition.

Cited references:

1. Badri, K. R. *et al.* Exonic Mutations of TSC2/TSC1 Are Common but Not Seen in All Sporadic Pulmonary Lymphangiomyomatosis. *Am. J. Respir. Crit. Care Med.* **187**, 663–665 (2013).
2. Ogorek, B. *et al.* Generalized mosaicism for TSC2 mutation in isolated Lymphangiomyomatosis. *European Respiratory Journal* 1900938 (2019)
DOI:10.1183/13993003.00938-2019.
3. Travaglini, K. J. *et al.* A molecular cell atlas of the human lung from single-cell RNA sequencing. <http://biorxiv.org/lookup/doi/10.1101/742320> (2019) DOI:10.1101/742320.
4. Reyfman, P. A. *et al.* Single-Cell Transcriptomic Analysis of Human Lung Provides Insights into the Pathobiology of Pulmonary Fibrosis. *Am J Respir Crit Care Med* **199**, 1517–1536 (2019).
5. McGinnis, C. S., Murrow, L. M. & Gartner, Z. J. DoubletFinder: Doublet Detection in Single-Cell RNA Sequencing Data Using Artificial Nearest Neighbors. *Cell Systems* **8**, 329-337.e4 (2019).
6. Gonzalez, R. F., Allen, L., Gonzales, L., Ballard, P. L. & Dobbs, L. G. HTII-280, a Biomarker Specific to the Apical Plasma Membrane of Human Lung Alveolar Type II Cells. *Journal of Histochemistry & Cytochemistry* **58**, 891–901 (2010).
7. Nabhan, A. N., Brownfield, D. G., Harbury, P. B., Krasnow, M. A. & Desai, T. J. Single-cell Wnt signaling niches maintain stemness of alveolar type 2 cells. *Science* **359**, 1118–1123 (2018).
8. Zacharias, W. J. *et al.* Regeneration of the lung alveolus by an evolutionarily conserved epithelial progenitor. *Nature* **555**, 251–255 (2018).

9. Zepp, J. A. *et al.* Distinct Mesenchymal Lineages and Niches Promote Epithelial Self-Renewal and Myofibrogenesis in the Lung. *Cell* **170**, 1134-1148.e10 (2017).
10. Barkauskas, C. E. *et al.* Type 2 alveolar cells are stem cells in adult lung. *J Clin Invest* **123**, (2013).

RESPONSE TO REVIEWER 2

We greatly appreciate the critical review by Reviewer#2 and encouraged by his/her opinion of finding our data “an exciting piece of work.” Accordingly, with the Reviewer’s valuable comments and suggestions, we performed new data analysis, clarified some of the experimental data, and provided additional information in the revised manuscript. Our responses to the Reviewer’s comments and point-by-point discussion are presented below.

“1. Authors identified LAM cells from parallel comparison of UMAP analysis on normal and patient scRNA-seq data. It might be originated from batch effect of two experiments from two different individuals. I ask authors to show cell type cluster data with aggregated samples from patient and normal lung together.”

RESPONSE: To address the Reviewer’s insightful and valuable comment, we performed data integration of the LAM and Control lungs scRNA-seq data. Presented below, is a side-by-side comparison of the dimensionality reduction plots (uMAPs) built for the integrated data. The left-side uMAP shows cellular subtypes discovered in the combined dataset, while the right-side uMAP demonstrates the source of the cells in each cluster (LAM or healthy control lung). Most of the cell clusters demonstrated the overlap, however, 100% of LAM lung cells (entire cluster 17) originates from the LAM lung.

scRNA-seq of human lungs, integrated data analysis

“2. Is there any possibility of cellular interaction between LAM or MANC with immune cells such as alveolar macrophages for the pathogenesis of LAM.”?

RESPONSE: We appreciate the Reviewer’s question and agree that the cellular interaction between LAM or MANCs (now termed Mesenchymal Alveolar Cells (MACs) with immune cells may contribute to the pathogenesis of LAM. When looking at the side-by-side comparison of the control and LAM lung scRNA-seq datasets, we noticed that control lung immune cells were represented by 2 macrophages clusters. In contrast, LAM lung immune cell populations additionally included mature T cells and B cells. Interestingly, when looking at the integrated data set (described above) we noticed a subset of macrophages, originating mostly from the LAM lung – cluster 15. Among the top differentially expressed genes in this cluster we highlighted the expression of the *CD86* gene, encoding T-Lymphocyte Activation Antigen CD86.

However, the focus of the current study was the interaction between mesenchymal and alveolar epithelial cells in human LAM lung and subsequently the creation of the genetic mouse model of LAM. We plan to address the interesting role of immune cells in the LAM lung and investigate the interaction of these cell types with LAM lung cells in future studies.

“3. Is there any difference in cell type proportion? For example, immune cell cluster of normal lung in Fig 1a looks bigger than that of patient lung. “

RESPONSE: The starting proportion of immune cells used in our scRNA-seq analysis was the same. To clarify our response, we would like to refer to the Methods section where we described the procedure of scRNA-seq sample preparation. Prior to single-cell capture, single-cell lung suspensions were depleted of the immune cells using human anti-CD45 MACS magnetic beads. This step was added to ensure the enrichment of the single-cell suspension with the cells of our interest – mesenchymal and epithelial subtypes. The total number of cells submitted for single-cell capture was determined as 20,000 cells. Before the capture, 10% (or 2,000) of separated CD45+ immune cells were added back to the samples (spiked). Therefore, at the moment of single-cell capture, both control and LAM lung samples were composed of 90% of CD45neg and 10% of original immune cells. However, the efficiency of capture for single-cell analysis in each sample may vary due to the limitation imposed by the technical variation of the method. Therefore, we don’t think it is appropriate to directly compare the number of cells in specific clusters between the samples.

“4. What is the immune cell cluster in Fig 1a? Authors used CD45-depleted cells in single-cell profiling, but still it is not clear.”

RESPONSE: As we already mentioned in our previous responses, immune cells were not completely absent in the RNA-seq analysis. After depletion of the samples for the CD45+ immune population, which turned out to constitute over 85% of all cells, we have added 10% of them back. That way we ensured a more equal representation of all cell types including the immune cells in the analysis.

In **Fig. 1a** there are two immune cell clusters which we identified as macrophages and “proliferative macrophages” based on their differential gene expression of macrophage cell markers (*CD68*, *CD169*, *CD206*), and proliferation markers (*PCNA*, *Ki67*).

“5. Authors used large cell numbers for single-cell analysis, but sequencing read number for single cell is relatively low. Is there any reason to use low sequencing depth?”

RESPONSE: We appreciate the Reviewer’s question. We are aware that the sequencing depth in our dataset is at the lower range. However, the sequence saturation of the libraries was > 50%, and according to the advisement of the 10xGenomic Support Team, we could not achieve greater sequencing depth for our samples in these conditions, while the current level is sufficient to cluster cell sub-populations as per our goal of the manuscript.

(<https://kb.10xgenomics.com/hc/en-us/articles/115002022743-What-is-the-recommended-sequencing-depth-for-Single-Cell-3-and-5-Gene-Expression-libraries->).

“6. In Fig 1e and 1f, authors described cellular interactions in LAM and normal lung. However, it was based on co-expression of receptor and ligand in LAM lung, so it requires to be investigated meticulous with validated methods such as CellPhoneDB.”

RESPONSE: To address the Reviewer’s criticism, we performed CellPhoneDB analysis of cellular interactions in control and LAM lungs. Although CellPhoneDB uses almost the same databases as FantomDB, which we used for the analysis in our manuscript, we noticed that the results were not a 100% match. This stems from the difference in the filtering cutoff threshold. CellphoneDB’s default cutoff for filtering based on expression is 10%. This means that if the genes in the interacting pair are expressed in more than 10% of the cells in the corresponding group, it is considered to be a ligand-receptor pair. In the previous analysis, we used a 30% cutoff.

However, when compared to the cellular crosstalk diagrams obtained with CellPhoneDB results against our previous analysis, we concluded that they were very similar.

In both applied methods the results were filtered out to keep only those pairs that were statistically significant (have a p-value of less than 0.05). In the revised manuscript we decided to keep our previous diagrams in **Figure 1g** since they are less busy and better highlight the contrast in signaling directionality. FantomDB¹ method for the analysis of cellular interactions was

statistically significant (have a p-value of less than 0.05). In the revised manuscript we decided to keep our previous diagrams in **Figure 1g** since they are less busy and better highlight the contrast in signaling directionality. FantomDB¹ method for the analysis of cellular interactions was

previously used in scRNA-seq analysis and the results were successfully published². However, to address the Reviewer's critique, we revised the text and figures of the manuscript and excluded the details of specific ligand-receptor pairs, discussed in the original submission.

"7. What is the role of TSC gene deletion in LAM cells? Please show the changes in mTORC signaling and status of TSC1/TSC2 in single cell of LAM patient."

RESPONSE: The patient's clinical history for the LAM lung samples used in our study, specifically the lack of family history of LAM or TS and records of the genotyping data for *TSC1/2* mutations, indicates a sporadic LAM. In sporadic pulmonary LAM, exonic mutations of *TSC1/TSC2* genes are not commonly seen³, or the mutations could be mosaic and not easily detectable⁴. Following the comparative analysis of the cell types in the LAM lung and control human lung, we have identified a unique cell subtype in the LAM lung which did not have an equivalent in the control healthy lung which we named "LAM lung cells". We are aware that the term "LAM cells" has been historically used to define TSC2-negative cells. However, we were unable to look for point mutations or to show the loss of *TSC1* or *TSC2* gene expression in the newly identified LAM lung cell subset, nor the other cell types in the LAM lung due to insufficient depth of the scRNA-seq method. Subsequently, we refrained from calling this cell subtype LAM cells, but instead, we named them "LAM lung cells" to emphasize its unique origin. We agree with the reviewer that it is important to provide clarification about the status of *TSC1/2* mutations. We have included substantial clarification of this matter in the discussion of the revised manuscript.

However, to at least partially address Reviewer's question, we compared the differentially expressed genes in "LAM lung cells" to the upregulated genes previously published in our study⁵ in the known TSC2-null human LAM cell line – kidney angiomyolipoma (AML) cells⁶ and found a number of genes in common (new **Extended Data Fig. 4D** in the revised manuscript). Many of these genes appeared to be related to the mTORC1 pathway up-regulation.

"8. In addition, LAM-specific marker can be validated in patient tissue. Visualization of LAM cells and increased population of LAM/MANC cells in patient tissue might provide more convincing evidence."

RESPONSE: We greatly appreciate the valuable suggestion by the Reviewer about visualization of LAM cells and the LAM/MAC population. In the **Extended Data Figure 1**, we showed IHC staining for pS6, a marker of mTORC1 activation. We also validated one of the markers of the newly discovered LAM lung cells – MYH10, stained in LAM lung lesions (on the right) as opposed to the control on the left (**Extended Data Figure 1C**). Additionally, we demonstrated the co-localization of MYH10 with smooth muscle α -actin, which is known to be expressed by LAM cells. We will follow up on the Reviewer's suggestions with more molecular markers of LAM lung cells validated in the patient tissue in our future studies.

Cited references:

1. Bono, H., Kasukawa, T., Furuno, M., Hayashizaki, Y. & Okazaki, Y. FANTOM DB: database of Functional Annotation of RIKEN Mouse cDNA Clones. *Nucleic Acids Res.* **30**, 116–118 (2002).
2. Zepp, J. A. *et al.* Distinct Mesenchymal Lineages and Niches Promote Epithelial Self-Renewal and Myofibrogenesis in the Lung. *Cell* **170**, 1134-1148.e10 (2017).
3. Badri, K. R. *et al.* Exonic Mutations of TSC2/TSC1 Are Common but Not Seen in All Sporadic Pulmonary Lymphangiomyomatosis. *Am. J. Respir. Crit. Care Med.* **187**, 663–665 (2013).
4. Ogorek, B. *et al.* Generalized mosaicism for TSC2 mutation in isolated Lymphangiomyomatosis. *European Respiratory Journal* 1900938 (2019)
doi:10.1183/13993003.00938-2019.
5. Himes, B. E. *et al.* Rapamycin-independent IGF2 expression in Tsc2-null mouse embryo fibroblasts and human lymphangiomyomatosis cells. *PLoS ONE* **13**, e0197105 (2018).
6. Yu, J., Astrinidis, A., Howard, S. & Henske, E. P. Estradiol and tamoxifen stimulate LAM-associated angiomyolipoma cell growth and activate both genomic and nongenomic signaling pathways. *Am J Physiol Lung Cell Mol Physiol* **286**, L694-700 (2004).

RESPONSE TO REVIEWER #3

We greatly appreciate the critical review and comments by the Reviewer. We were delighted by the opinion of the Reviewer that our study “elegantly developed a novel mouse model of LAM sheds light on the origin of LAM.” In response to the valuable suggestions of the Reviewer, we performed additional new experiments to provide a deeper understanding of the mechanism and to address his/her concerns to improve our manuscript. We included new data in extensively revised Figures and manuscript. A point-by-point response explaining how we have addressed each of the Reviewer’s comments is described below.

Major comments:

“1. Rapamycin had a partial inhibitory effect in the 3D organoid from TSC-KO mesenchymal cells (Figures 3g and h), and rapamycin withdrawal led to the increase in the size of the 3D organoid colonies. It would be interesting to see if pS6, SMA, MYH10 also decreased with rapamycin and increased with rapamycin withdrawal. This will help demonstrate that mTOR activation is the key pathway in this system.”

RESPONSE: To address the Reviewer’s valuable suggestion, we performed additional experiments to demonstrate the effect of rapamycin on pS6, SMA, MYH10. We used freshly isolated *Tsc2*^{WT} and *Tsc2*^{KO} mesenchymal lung cells following the magnetic sort procedure schematically shown in the **Extended Figure 6A**. *Tsc2*^{WT} and *Tsc2*^{KO} mesenchymal lung cells were cultured in 3D Matrigel with or without 20nM rapamycin. Cells then were cleared out of the Matrigel matrix using a Cell recovery solution, washed with FACS buffer, and used for mRNA extraction. For this experiment, we used fibroblasts isolated from 3 biological replicates of each 8-week-old *Tbx4*^{LME-Cre}*Tsc2*^{WT} and *Tbx4*^{LME-Cre}*Tsc2*^{KO} mouse lungs in each condition.

Figure 1. Effect of rapamycin on mTORC1 activation (A) and on SMA and MYH10 expression (B).

We used Western Blot analysis to verify phosphorylation levels of S6. As seen in figure 1A, pS6 a marker of mTORC1 activation¹, was markedly increased in serum-deprived *Tsc2*^{KO} mesenchymal lung cells further confirming that *Tsc2*^{KO} induces the constitutive mTORC1 activation compared to very little basal levels of pS6 in *Tsc2*^{WT} mesenchymal lung cells. Importantly, treatment with 20nM rapamycin decreased pS6. Fig. 1B shows qRT-PCR results demonstrating that the gene expression of *Acta2* and *Myh10* genes were not affected by rapamycin treatment. Collectively, these data suggest that mTORC1 activation (pS6 protein level) is affected by rapamycin while the gene expression of *ACTA2* and *MYH10*, encoding SMA and MYH10, was not. However, it is known that the mTOR pathway controls protein translation, and we may not exclude the possibility of

the SMA and MYH10 protein translation inhibition by rapamycin. We will address this question and the effect of rapamycin withdrawal suggested by the Reviewer in our future studies.

“2. In this paper, the authors stated that the LAM phenotype with TSC2 KO in lung mesenchymal cells is correlated with the upregulation of the WNT signaling pathway, (WNT ligands, Figure 4c). To show this as a mechanism, it would be beneficial to see if increased expression of WNT ligands in normal lung mesenchymal cells would develop into the LAM phenotype. Also, if rapamycin inhibits the up-regulation of WNT ligands.”

RESPONSE: To address the Reviewer’s insightful comment, we completed *in vivo* experiments, performed new immunohistochemical, and morphometric analysis. To determine whether the upregulation of WNT signaling would develop into the LAM phenotype, we crossed *Tbx4*^{LME-Cre} mice with *Ctnnb1*^{Ex3floxed} mice² to create *Tbx4*^{LME-Cre}*Ctnnb1*^{Ex3KO} mice with stable expression of β -catenin in lung mesenchyme. Stabilization of β -catenin is achieved through the KO of the Exon3 of the β -catenin gene *Ctnnb1*. Such modification preserves the functional structure of the protein, while it eliminates the site for phosphorylation, which targets β -catenin for degradation in the absence of WNT ligands leading to the constitutive activation of WNT signaling pathway. The mouse genotype description is schematically pictured in the new **Extended Data Fig. 5** of the revised manuscript.

Alveolar size assessment of the 20-week-old *Tbx4*^{LME-Cre}*Ctnnb1*^{Ex3KO} mouse lungs demonstrated that mean linear intercept (MLI) measurements for *Tbx4*^{LME-Cre}*Ctnnb1*^{Ex3KO} mice were comparable to WT control. This suggests that the upregulation of the WNT signaling pathway in lung mesenchymal cells is not sufficient to induce the LAM lung phenotype (alveolar enlargement). New data is included in the new **Fig. 2h** of the revised manuscript.

To address the Reviewer’s question “if rapamycin inhibits the upregulation of Wnt ligands, we performed new additional *in vivo* experiments. Briefly, 4-week-old female *Tbx4*^{LME-Cre}*Tsc2*^{KO} and *Tbx4*^{LME-Cre}*Tsc2*^{WT} mice were treated with rapamycin by i.p. injections (4mg/kg, 3x week) for 4 weeks. *Tsc2*^{WT} and *Tsc2*^{KO} mesenchymal lung cells were isolated from 8-week-old mouse lungs using the magnetic sort procedure as schematically shown in **Extended Data Fig. 6A**. Sorted mesenchymal cell fractions were used for mRNA extraction and gene expression analysis by qRT-PCR. In the original manuscript, we demonstrated the results of the popRNA-seq and qRT-PCR analysis of *Wnt3a*, *Wnt5a* upregulation in *Tsc2*^{KO} mesenchymal lung cells of the 8-week-old female mice. (**Fig. 2e** and **Extended Data Fig. 7B**). In the new experiment, we found that *in vivo* treatment with rapamycin significantly inhibits *Wnt3a* and *Wnt5a* gene expression in *Tsc2*^{KO} mesenchyme of the 8-week-old female mice. This evidence suggests that the upregulation of *Wnt3a* and *Wnt5a* in *Tsc2*^{KO} mouse lung mesenchyme is mTORC1-dependent. The new data is included in the revised text of the manuscript and the new **Fig. 2f**.

“3. The paper interestingly shows that mTOR activation due to TSC2 loss activates ER and its target genes (extended Figure 7A-C), which is a very novel finding. They showed that gender-specific and age-specific LAM phenotype with a decline in LAM-lung function more in females,

more so in breeders. They also showed that mTOR activation due to TSC2 loss increases WNT ligands in lung mesenchyme that activates WNT signaling pathway. However, there is a lack of mechanistic connection between mTOR, WNT signaling, and ER pathways with the LAM phenotype.”

RESPONSE: We appreciate the Reviewer’s comment regarding “mechanistic connection between mTOR, WNT signaling and ER pathways and LAM phenotype.” To address the Reviewer’s comment, we performed an additional *in vivo* experiment to demonstrate a mechanistic connection between mTOR, WNT signaling, and sex. The experiment is described in the previous response.

Among the transcriptomic changes caused by the loss of *Tsc2* in lung mesenchyme, we discovered the upregulation of several important molecular pathways including WNT (**Extended data Fig. 6C** of the revised manuscript) which could affect the observed phenotype. Moreover, we have found some of the key genes in the WNT pathway to be regulated in a sex-specific manner (**Extended Data Figs. 7A and 7B** of the revised manuscript), suggesting a potential involvement of estrogen transcription regulation. In agreement with that, we have found that *Tsc2*^{KO} MLFs had increased expression of the *Esr1* gene, encoding ER α (**Extended Data Fig. 7D** in the revised manuscript). It has been previously shown that the ligand-independent transactivation of the ER α specifically depends on S167 phosphorylation performed by S6K1³, an enzyme that is hyperactive in TSC2-null cells, suggesting that it could lead to synergistic upregulation of ER α gene targets by simultaneous action of high estrogen and constitutively active mTORC1 pathway. This could explain the increased expression of selected genes (e.g., *Wnt3a*, *Wnt5a*, etc.) specifically in female *Tsc2*^{KO} MLFs lungs and not female *Tsc2*^{WT} MLFs. The hypothetical scheme of the three pathways interplay is presented in the **Extended Data Fig. 10** of the revised manuscript.

To further address the valuable Reviewer’s suggestion, we plan to continue investigating the mechanistic connection between mTOR, WNT, and ER pathways. We will determine whether estrogen depletion in female ovariectomized *Tbx4*^{LME-Cre}*Tsc2*^{KO} mice will affect WNT ligands expression in *Tsc2*^{KO} lung mesenchymal cells. In parallel, we will perform additional experiments using a combination of estrogen depletion and rapamycin treatment. Further, to demonstrate the role of WNT/ β -catenin signaling in the development of the LAM phenotype we will use WNT inhibitors, e.g. ICG-001, a selective blocker of the interaction between β -catenin and the transcriptional co-activator CREB-binding protein (CBP). These studies will provide valuable preclinical data for the novel treatment of LAM. Because of the Editorial time constraints for the resubmission of the revised manuscript, and because *in vivo* experiments described above require extended time, we will follow up on the Reviewer’s suggestion in our future studies.

We also included the following text in the discussion section of the revised manuscript: “Collectively, based on the published data and our novel findings we created a hypothetical scheme of the intricate molecular interplay of the three pathways: mTORC1, ER α and WNT in the development of LAM-like phenotype in our mouse model (**Extended Data Fig. 10**). Due to the high degree of complexity of all three molecular pathways involved, the more detailed mechanism will have to be elucidated in our following studies.”

“4) Extended Fig 2B: Since AT2 signature genes are expressed in the AT1 gene clusters and vice versa in the control lungs, the category saying AT1/AT2 needs a little more definition, like how many-fold increase in both AT1 and AT2 genes. (Same for Fig 1 d).”

RESPONSE: We are not sure how to address the Reviewer’s question because of the limitations of the scRNA-seq analysis - this experimental approach does not allow us to present the gene expression data as a fold change to demonstrate biologically meaningful information⁴⁻⁶. When compared gene expression of one cell cluster to another, we can devise an average log fold change in gene expression across all cells in the cluster. Meanwhile, it is not always the case that the gene is expressed in all cells in the cluster and it is not expressed at the same levels. Thus, averaging the expression across the population and then comparing it to another average does not reflect the true changes on a cell by cell level. However, to address the Reviewer’s comment we have presented the results of such comparisons in the table below.

We understand the Reviewer’s skepticism regarding the biological truth behind the AT2/AT1 transitional state cell cluster. To prove the existence of the transitional AT2/AT1 cell type *in vivo*, we performed double immunofluorescent staining of lung sections from multiple control and LAM lungs using cell-type-specific antibodies – HTII-280 for AT2 cells and HTI-56 to detect AT1 cells⁷. Quantitative analysis of the series of confocal images identified approximately 3.5% of total alveolar epithelial cells (AECs) in control lung to be double-positive for AT2 and AT1 markers. In contrast, LAM lungs showed on average 17.5% of AECs to be AT1/AT2 transitional cells. These new data are displayed in the revised **Fig. 1, panels e and f**, and described in detail in the revised text of the manuscript.

Differential comparison of AT1 cell marker gene expression in AT2/AT1 cells cluster compared to AT1 cluster

Gene	P-value	Average_logFC	% of cells in the cluster expressing the gene	
			AT1 cell cluster	AT2/AT1 cell cluster
AGER	1.77E-31	1.06	99.3	100
HOPX	2.11E-16	0.57	97	100
AQP4	0.000725	0.56	71	93.2

Differential comparison of AT2 cell marker gene expression in AT2/AT1 cells cluster compared to AT2 cluster

Gene	P-value	Average_logFC	% of cells in the cluster expressing the gene	
			AT2 cell cluster	AT2/AT1 cell cluster
SFTPC	4.33E-22	0.45	100	100
SFTPA1	4.95E-12	0.39	99.8	100
SFTPB	1.05E-15	0.28	100	100

Differential comparison of AT1 cell marker gene expression in AT1 cells cluster compared to AT2 cluster

Gene	P-value	Average_logFC	% of cells in the cluster expressing the gene	
			AT2 cell cluster	AT2/AT1 cell cluster
AGER	0	4.23	99.3	30.3
HOPX	0	1.10	97	98.5
AQP4	2.53E-135	1.32	71.1	53.5

Differential comparison of AT2 cell marker gene expression in AT2 cells cluster compared to AT1 cluster

Gene	P-value	Average_logFC	% of cells in the cluster expressing the gene	
			AT2 cell cluster	AT2/AT1 cell cluster
SFTPC	0	4.31	100	96.9
SFTPA1	0	4.07	99.8	35.3
SFTPB	0	1.32	100	87.2

“5) Figure 2d: This figure shows that WNT signaling is increased by TSC2 KD in lung mesenchymal cells, shown in the next figure by the increase of WNT ligands. Further experiments showing the importance of WNT signaling pathways as an important pathway in LAM would be helpful. For example, experiments showing increased expression of WNT ligands in normal lung mesenchyme lead to the LAM phenotype. Or, experiments showing that inhibition of WNT signaling pathways reverses the LAM phenotype.”

We appreciate the Reviewer’s comment raising an important question about the role of the WNT signaling pathway in the development of the LAM phenotype. To address this insightful and valuable suggestion, we performed additional experiments and presented new data analysis in Figure 2g of the revised manuscript.

To test whether the inhibition of the WNT signaling pathway reverses the LAM phenotype, we crossed *Tbx4*^{LME-Cre}*Tsc2*^{floxed} mice with *Ctnnb1*^{(Ex2-6)-floxed} mice⁸, creating a double conditional KO of *Tsc2* and β -catenin (*Ctnnb1*) genes in lung mesenchymal cells - *Tbx4*^{LME-Cre}*Tsc2*^{KO}*Ctnnb1*^{KO} mice (**Extended Data Fig. 5** of the revised manuscript). Subsequent analysis of the lung phenotype in these mice revealed that the addition of WNT pathway inhibition to the *Tsc2*-dependent mTORC1 activation in lung mesenchyme exerts a significant age-dependent protective effect, as evident from gradual MLI decrease in the lungs of 8-, 16- and 20-week-old mice. (**Fig. 2g** of the revised manuscript). In fact, at the age of 20 weeks the size of the alveoli was restored to the WT control levels. This observation underscored the important role of the molecular interplay of the WNT and mTORC1 pathways in the development of the alveolar enlargement phenotype in *Tbx4*^{LME-Cre}*Tsc2*^{KO} mouse lungs and was included in the revised manuscript.

As we have already reported in the response to comment #2, mTORC1-independent activation of WNT in the lung mesenchyme (through stabilization of β -catenin in *Tbx4*^{LME-Cre}*Ctnnb1*^{Ex3KO} mice, **Extended Data Fig. 5** of the revised manuscript), did not trigger the alveolar enlargement (**Fig. 2h** of the revised manuscript), demonstrating that activation of WNT signaling alone in lung mesenchyme is not sufficient to induce LAM lung phenotype.

Minor comments:

"1) Fig. 1: They don't say how many subjects were part of this?"

RESPONSE: To address the Reviewer's comment, we revised a legend to Figure 1a, as follows: "UMAP dimensionality reduction plots representing scRNA-seq analysis of the lungs from the LAM patient (2 samples) and age- and sex-matched control donor (2 samples)."

"2) Fig. 2a: Tsc2 expression is not quite as gone in the KO compared to the whole lungs?"

RESPONSE: Western blot analysis was performed on the whole lungs as well on the total population of mouse lung fibroblasts (MLFs) sorted according to the scheme shown in **Extended Data Fig. 6A**. In the whole *Tbx4*^{LME-Cre}*Tsc2*^{KO} lungs, we expected to see some TSC2 expression, since the KO is targeted only to the TBX4+ mesenchymal lineage. In the sorted *Tsc2*^{KO} mouse lung mesenchyme, we still detect, although at a much lower level, the expression of TSC2. This may be explained by either incomplete efficiency of Cre recombinase or by that (CD45^{neg}EpCAM^{neg}CD31^{neg}PDPN^{neg}) cell population may contain other, not TBX4-derived mesenchymal cells in which *Tsc2* was not deleted by Cre. However, as evident from the Western blot and immunofluorescent staining in **Figs. 2a and 2b**, this level of TSC2 KO in the lung mesenchyme was sufficient to trigger subsequent activation of mTORC1 signaling pathway.

"3) Figure 3A: the genes described in the text (lines 167-168) refer to Fig 3A, which do not show the genes Wnt3a, Wnt4, Wnt5a, and Wnt10b. although qPCR results are shown in extended Fig 7)."

RESPONSE: We appreciate the Reviewer's comment and have adjusted our figure panels (**Fig. 2e**, and **Extended Data Figs. 7A and 7B**) accordingly in the revised manuscript.

"4) Figure 3b: gene Fzd2 is not shown in the array although mentioned in the text (line184)".

RESPONSE: We appreciate the Reviewer's comment and have adjusted our figure panels (**Fig. 2e**, and **Extended Data Fig. 7A**) accordingly in the revised manuscript.

"5) Fig. 3d, g, h, j: how many mice/experiments used/performed?"

RESPONSE: Requested corrections were included in the figure legend of the revised **Fig. 3**.

"6) Extended Fig. 1: please spell out MYH-10 (myosin heavy chain-10) in the legend."

RESPONSE: We appreciate the Reviewer's suggestion and have included the full name for MYH10 in the figure legend of the **Extended Data Figure 1** of the revised manuscript.

"7) Extended Fig 5A: What does B, V, and B stand for? It's not written in the figure legend."

RESPONSE: Upon revision, we have excluded the aforementioned data from the revised manuscript.

"8) Extended Fig. 6B: number of mice used within each group not stated."

RESPONSE: Requested corrections were included in the figure legend of the revised **Extended Data Fig. 6B**.

"9) Extended Fig. 7A: What is the p-value comparison between the male and female knockouts in all of these groups as related to upregulation in ERalpha regulation to underscore sex as a biological variable?"

RESPONSE: To address the Reviewer's criticism, we included suggested statistical information in the revised **Extended Data Fig. 7B**.

"10) The overall organization of the manuscript into many extended figures that are referred to in the text out of order is very confusing. The authors may consider re-organizing the presentation of their data in a progressing order."

RESPONSE: To address the Reviewer's criticism, we extensively revised the text and figures.

"11) The Discussion is very descriptive, many of the details mentioned belong in the Introduction. It would have been better if the authors placed their findings in the context of the disease, focused on the interpretation of the results, and speculated on the underlying mechanisms behind their

observations. Specifically, it would be interesting to discuss future direction for this TSC2 model, as well as what new therapies are a possibility based on the findings.”

RESPONSE: To address the Reviewer’s criticism, we significantly revised the text in the introduction and re-wrote the discussion section to include the interpretation of the results.

Cited references:

1. Goncharova, E. A. *et al.* Tuberin regulates p70 S6 kinase activation and ribosomal protein S6 phosphorylation: a role for the TSC2 tumor suppressor gene in pulmonary lymphangiomyomatosis. *J. Biol. Chem.* **277**, 30958–30967 (2002).
2. Harada, N. Intestinal polyposis in mice with a dominant stable mutation of the beta -catenin gene. *The EMBO Journal* **18**, 5931–5942 (1999).
3. Yamnik, R. L. & Holz, M. K. mTOR/S6K1 and MAPK/RSK signaling pathways coordinately regulate estrogen receptor α serine 167 phosphorylation. *FEBS Letters* **584**, 124–128 (2010).
4. Travaglini, K. J. *et al.* A molecular cell atlas of the human lung from single cell RNA sequencing. <http://biorxiv.org/lookup/doi/10.1101/742320> (2019) doi:10.1101/742320.
5. Reyfman, P. A. *et al.* Single-Cell Transcriptomic Analysis of Human Lung Provides Insights into the Pathobiology of Pulmonary Fibrosis. *Am J Respir Crit Care Med* **199**, 1517–1536 (2019).
6. McGinnis, C. S., Murrow, L. M. & Gartner, Z. J. DoubletFinder: Doublet Detection in Single-Cell RNA Sequencing Data Using Artificial Nearest Neighbors. *Cell Systems* **8**, 329-337.e4 (2019).
7. Gonzalez, R. F., Allen, L., Gonzales, L., Ballard, P. L. & Dobbs, L. G. HTII-280, a Biomarker Specific to the Apical Plasma Membrane of Human Lung Alveolar Type II Cells. *Journal of Histochemistry & Cytochemistry* **58**, 891–901 (2010).
8. Brault, V. *et al.* Inactivation of the β -catenin gene by Wnt1-Cre-mediated deletion results in dramatic brain malformation and failure of craniofacial development. **12**.

REVIEWER COMMENTS

Reviewer #1 (Remarks to the Author):

The authors have addressed most of my concerns but I have a few remaining comments

1. Why can't the authors sequence the patient's genomic DNA for mutations in TSC1/TSC2? This should be possible from the samples provided (even from sections of the lungs)
2. Nomenclature of alveolar fibroblasts. The authors now use the nomenclature "Mesenchymal Alveolar cells" or "MACs" instead of MANCs. However, they conclude that these cells are similar to the category "alveolar fibroblasts" characterized by others. So why not use this unifying nomenclature instead? Clearly this is a broader problem for the field, but devising yet another terminology is surely only compounding the problem?
3. Preparation of lung samples prior to single cell analysis. This is now described (not very clearly) on page 5, but not in the Materials and Methods where it should be added. If the "blood vessels" are dissected away why does the "interactome" map for control lung in Fig 1g center on "macro vessels" and still not include interactions between stroma/MACs and AT2 cells? What are "macro vessels"? How reliable are the programs for revealing biologically relevant cell-cell interactions?
4. The authors now argue that the AT2 cells that express markers of AT1 cells are in a "transitional" state and "potentially active in alveolar repair". But they do not address my question as to how these cells compare transcriptionally with "transitional" AT2 cells described by others in lung repair/regrowth (Wu et al Cell. 2020 Jan 9;180(1):107-121.e17. doi: 10.1016/j.cell.2019.11.027 and Kobayashi et al doi: <https://doi.org/10.1101/855155> (now in press) Some discussion would be useful for the general reader.
5. Organoid assay. The authors need to state more clearly in the Material and Methods how the mesenchymal cells were isolated? Is this the same as in the scheme shown in extended data Fig 6? If so, it should be stated clearly

Reviewer #2 (Remarks to the Author):

All the questions were answered clearly. It will be interesting to understand the role of immune cells in the pathogenesis of LAM. I would recommend to utilize public or published single cell RNA-seq dataset to enhance the quality of this study. In addition, spatial transcriptome analysis might provide insight on LAM pathogenesis.

Reviewer #3 (Remarks to the Author):

The authors addressed my concerns.

Response to Reviewer #1:

As recommended by Reviewer #1, we substantially revised the original manuscript and explained the data based on his/her concerns.

Point-by-point responses on the few remaining comments raised by Reviewer #1 are fully addressed below.

“Q1. Why can’t the authors sequence the patient’s genomic DNA for mutations in TSC1/TSC2? This should be possible from the samples provided (even from sections of the lungs).”

Response: Indeed, sequencing of genomic DNA is possible from sections of the lung as demonstrated by recently published studies with a focus on detecting mutations of *TSC1* and/or *TSC2* in LAM lung. However, we were surprised by the Reviewer’s additional question and request. As we responded to the original question of the Reviewer, the patient’s clinical history indicates a diagnosis of sporadic LAM. To point this out we have added a sentence in the Discussion section of the revised manuscript (**Page 12**). In sporadic pulmonary LAM, exonic mutations of *TSC1/TSC2* genes are not commonly seen¹, or the mutations could be mosaic and not easily detectable². While inactivating mutations of *TSC1/TSC2* are associated with LAM disease, it is well established in a field that in many cases *TSC1* or *TSC2* mutations are not detectable in sporadic LAM lung¹. The exact *TSC1/TSC2* mutations in sporadic LAM does not affect the results or conclusions in our study.

“Q2. Nomenclature of alveolar fibroblasts. The authors now use the nomenclature “Mesenchymal Alveolar cells” or “MACs” instead of MANCs. However, they conclude that these cells are similar to the category “alveolar fibroblasts” characterized by others. So why not use this unifying nomenclature instead? Clearly this is a broader problem for the field, but devising yet another terminology is surely only compounding the problem?”

Response: In our original response to Reviewer #1, as requested by the Reviewer, we changed the name of this new cell population from “Mesenchymal Alveolar Niche Cells” to “Mesenchymal Alveolar Cells.” Furthermore, we have already addressed the reviewer’s request and concerns very thoroughly by comparing the gene expression patterns of the mesenchymal subpopulation discovered in our study to the mesenchymal subpopulation characterized by others. Additionally, we carefully revised the whole manuscript, Figures, and Figure legends to accommodate the original request of Reviewer #1.

We agree with the Reviewer that a broader problem exists for the field and there is no unifying nomenclature for alveolar mesenchymal cells. Our subset of “mesenchymal alveolar cells” is not identical to “alveolar fibroblasts” so, in our opinion, mesenchymal alveolar cells represent an appropriate definition for the population of cells in our study.

“Q3. Preparation of lung samples prior single-cell analysis. This is now described (not very clearly) on page 5, but not in the Materials and Methods where it should be added. If the “blood vessels”

are dissected away why does the “interactome” map for control lung in Fig 1g center on “macro vessels” and still not include interactions between stroma/MACs and AT2 cells? What are “macrovessels”? How reliable are the programs for revealing biologically relevant cell-cell interactions?”

Response: To further address the Reviewer’s comment, we additionally revised the text of the Materials and Methods section, where we clearly stated that the large vessels (arteries and arterioles), and bronchi were dissected out before scRNA-seq analysis. This was done to eliminate the majority of tissue containing smooth-muscle cells which are abundant in the large vessels and bronchi and to detect abnormal SMCs infiltrating the parenchyma of LAM lungs.

Thus, remaining venules, small blood vessels, and capillaries were included in scRNA-seq analyses. Macro- and micro-vascular endothelial cells which are depicted in the interactome maps were identified by scRNA-seq analysis using published endothelial gene expression markers³.

Further, as we previously responded to the Reviewer’s question, we identified a number of interactions predicted for MACs → AT2 axis both in the control and LAM lungs based on FantomDB ligand-receptor analysis. However, in the control lung, the number of interactions on this axis is substantially lower and were filtered out in the process of creating the diagrams shown in **Figure 1e (revised Figure 1g)**. These diagrams were meant to show the contrast in the cellular crosstalk patterns; therefore, they depict only the most robust interactions in each sample.”

We respectfully have to note that we don’t use the term “macrovessels” in our text, however, we do mention the dissection of “large vessels” and we use the term “macrovascular endothelial cells” in our scRNA-seq analysis. By the “large vessels, we meant visible by eye pulmonary arteries and arterioles. “Macrovascular endothelial cells” as identified in Neithamer et al.³ by IHC and scRNA-seq can be both arterial and venous origin and are characterized by high expression of *VWF* and *VCAM1* genes. We have expanded the explanation of these terms in the appropriate Methods Sections (**Page 19**).

The programs for analysis of the cell-cell interactions are based on hypothetical/predicted interactions of the known ligand-receptor pairs, described in the literature, and deposited in the bioinformatic database. However, the interactome maps, that we present in our scRNA-seq analysis are based on real gene expression data for each of the ligands and receptors in the predicted pairs. To be extra thorough, we used two different programs for building the cell-cell interactome maps: Phantom DB and CellPhone DB, which gave us similar results. We acknowledge that specific ligand-receptor pairs need to be verified by biochemical, protein-based approaches, but this is beyond the scope of our study. Our goal was to show the contrast in cellular crosstalk directionality.

“Q4. The authors now argue that the AT2 cells that express markers of AT1 cells are in a “transitional” state and “potentially active in alveolar repair”. But they do not address my question as to how these cells compare transcriptionally with “ transitional” AT2 cells described

by others in lung repair/regrowth (Wu et al Cell. 2020 Jan 9;180(1):107-121.e17. doi: 10.1016/j.cell.2019.11.027 and Kobayashi et al doi: <https://doi.org/10.1101/855155> (now in press). Some discussion would be useful for the general reader.”

Response: In the extensively revised manuscript we presented evidence that AT2/AT1 represent a “transitional” cell population as was originally requested by Reviewer #1. We have addressed the Reviewer’s #1 question about AT2/AT1 cells very thoroughly, performed additional bioinformatics analysis, and new immunofluorescent staining in human lung sections to provide the biological confirmation of the presence of the AT2/AT1 transitional state at both the mRNA and the protein level.

In this new request, Reviewer #1 argues that we need to compare our data to very recently published papers on AT2 to AT1 plasticity. Reviewer #1 further requests the transcriptional comparison of our transitional AT2/AT1 cells to the “Pre-alveolar type-1 transitional cell state” (PATS) cells⁴ in these newly published studies. We followed up on this new request and found similarities in their unique transcriptional signatures of PATS and the AT2/At1 transitional cells discovered in LAM lung (e.g. the expression of *LGALS*, *SFN*, *CLDN4*, and *KRT8*, which were defining PATS cells). However, in our scRNA-seq analysis we have found that *SFN*, *CLNDN4*, and *KRT8* genes were not exclusively expressed in the alveolar epithelial cells. We included this additional new comparison in the Discussion section (**Page 12**) and new references in the Bibliography of the re-revised manuscript.

“Q5. Organoid assay. The authors need to state more clearly in the Material and Methods how the mesenchymal cells were isolated? Is this the same as in the scheme shown in extended data Fig 6? If so, it should be stated clearly.”

Response: To address the Reviewer’s comment, we have now stated more clearly in the Material and Methods (**Page 21-22**) how the mesenchymal cells were isolated for the organoid assays: they were isolated as shown in the scheme shown in Extended Data Fig 6.

1. Badri, K. R. *et al.* Exonic Mutations of TSC2/TSC1 Are Common but Not Seen in All Sporadic Pulmonary Lymphangiomyomatosis. *Am. J. Respir. Crit. Care Med.* **187**, 663–665 (2013).
2. Ogorek, B. *et al.* Generalized mosaicism for TSC2 mutation in isolated Lymphangiomyomatosis. *Eur. Respir. J.* 1900938 (2019) doi:10.1183/13993003.00938-2019.
3. Niethamer, T. K. *et al.* Defining the role of pulmonary endothelial cell heterogeneity in the response to acute lung injury. *eLife* **9**, e53072 (2020).
4. Kobayashi, Y. *et al.* Persistence of a novel regeneration-associated transitional cell state in pulmonary fibrosis. <http://biorxiv.org/lookup/doi/10.1101/855155> (2019) doi:10.1101/855155.

REVIEWERS' COMMENTS:

Reviewer #1 (Remarks to the Author):

The manuscript has been improved by the clarifications